# Structural basis for adhesion G protein-coupled receptor Gpr126 function

Katherine Leon[1,2], Rebecca L. Cunningham[3], Joshua A. Riback[1,4], Ezra Feldman[1], Jingxian Li[1,2], Tobin R. Sosnick[1,5], Minglei Zhao[1], Kelly R. Monk[3,6] & Demet Araç [1,2]*

Many drugs target the extracellular regions (ECRs) of cell-surface receptors. The large and alternatively-spliced ECRs of adhesion G protein-coupled receptors (aGPCRs) have key functions in diverse biological processes including neurodevelopment, embryogenesis, and tumorigenesis. However, their structures and mechanisms of action remain unclear, hampering drug development. The aGPCR Gpr126/Adgrg6 regulates Schwann cell myelination, ear canal formation, and heart development; and *GPR126* mutations cause myelination defects in human. Here, we determine the structure of the complete zebrafish Gpr126 ECR and reveal five domains including a previously unknown domain. Strikingly, the Gpr126 ECR adopts a closed conformation that is stabilized by an alternatively spliced linker and a conserved calcium-binding site. Alternative splicing regulates ECR conformation and receptor signaling, while mutagenesis of the calcium-binding site abolishes Gpr126 function in vivo. These results demonstrate that Gpr126 ECR utilizes a multi-faceted dynamic approach to regulate receptor function and provide relevant insights for ECR-targeted drug design.

[1] Department of Biochemistry and Molecular Biology, The University of Chicago, Chicago, IL 60637, USA. [2] Grossman Institute for Neuroscience, Quantitative Biology and Human Behavior, The University of Chicago, Chicago, IL 60637, USA. [3] Department of Developmental Biology, Washington University School of Medicine, St. Louis, MO 63110, USA. [4] Graduate Program in Biophysical Sciences Program, The University of Chicago, Chicago, IL 60637, USA. [5] Institute for Biophysical Dynamics, The University of Chicago, Chicago, IL 60637, USA. [6] Vollum Institute, Oregon Health & Science University, Portland, OR 97239, USA. *email: arac@uchicago.edu

Multicellular organisms rely on cellular communication to carry out critical biological processes, and numerous cell-surface receptors utilize their extracellular regions (ECRs) to modulate these cellular-adhesion and signaling events. For example, the ECRs of integrins, epidermal growth factor receptor (EGFR), and several G protein-coupled receptors (GPCRs) change conformation upon ligand binding, which propagates signals across the membrane[1–9]. Targeting the essential ECRs of receptors with antibody-like drugs to trap the ECRs in distinct conformations, or to modulate ECR-ligand interactions has been an effective way to treat diseases caused by defective proteins. Currently, the anti-cancer drug cetuximab targets EGFR to prevent an activating extended ECR conformation[10], and the drug etrolizumab blocks ligand binding to the ECRs of integrins in order to treat inflammatory bowel diseases[11]. Remarkably, earlier this year the migraine preventive drug erenumab, which blocks ligand binding to the ECR of calcitonin receptor-like receptor, became the first antibody drug against a GPCR to be approved by the Food and Drug Administration[12,13]. Despite these and other breakthroughs, there are many essential receptors in the human genome that are not currently drugged, including the 32 adhesion GPCRs (aGPCRs), a diverse and understudied family of GPCRs with critical roles in synapse formation, angiogenesis, neutrophil activation, embryogenesis, and more[14–16].

Like all GPCRs, aGPCRs have canonical signaling seven-transmembrane (7TM) domains[17,18]. However, unlike most other GPCRs, aGPCRs have large ECRs, which can extend up to almost 6000 amino acids (aa) and consist of various adhesion domains that mediate cell–cell and cell–matrix interactions[19]. In addition, during biosynthesis, aGPCRs are uniquely autoproteolysed within a conserved GPCR Autoproteolysis INducing (GAIN) domain of the ECR that is juxtaposed to the 7TM[20], resulting in a fractured receptor that nevertheless remains tightly associated at the cell surface[21,22].

Although their protein architectures remain largely unknown, functional studies have shown that aGPCR ECRs can regulate receptor function and that antibody-like synthetic proteins that target the ECRs can modulate downstream signaling[9,22–25]. A current model for aGPCR regulation suggests that transient interactions between the ECR and 7TM directly regulate receptor signaling[9,22–25]. There are also numerous reports that aGPCRs use their ECRs to mediate functions in a 7TM-independent manner[26–30]. Another non-mutually exclusive model for aGPCR activation posits that ligand binding to the ECR can exert force and cause dissociation at the autoproteolysis site, revealing a tethered peptide agonist, which then activates the receptor[31–34]. Clearly, the ECRs of aGPCRs have significant and diverse roles but remain poorly understood at a molecular level due to the scarcity of structural information, such as interdomain interactions, protein architecture, and identities of extracellular domains, which would provide insight into their mechanisms of action.

Gpr126/Adgrg6 is one of the better studied aGPCRs and is essential for Schwann cell (SC) myelination and other functions[35–37]. In vertebrate peripheral nervous system (PNS) development, the myelin sheath surrounding axons is formed by SCs and functions to facilitate rapid propagation of action potentials[38]. Disruption of myelination is associated with disorders such as Charcot-Marie-Tooth disease, which is characterized by muscle weakness[39,40]. In gpr126-mutant zebrafish, SCs fail to express genes critical for myelination during development and are not able to myelinate axons due to deficient G-protein signaling. Additional studies have shown that this regulatory function of Gpr126 is conserved in mammals[41,42] and that Gpr126 also plays a role in myelin maintenance through communication with the cellular prion protein[43]. In humans, GPR126 mutations are linked to several cancers and other diseases[44–48], including adolescent idiopathic scoliosis[49] and arthrogryposis multiplex congenita, a disorder characterized by multiple joint contractures[50]. Furthermore, Gpr126 is required for inner ear development in zebrafish[35] and GPR126 is required for heart development in mouse[37], and it has been shown that the latter function is ECR-dependent and does not require the 7TM[30]. While the biological significance of Gpr126 has become indisputable over recent years, the molecular mechanisms underlying Gpr126 functions remain unclear.

Gpr126 has a large ECR consisting of 839 aa. Prior to the current study, four domains in the ECR of Gpr126 had been identified through sequence-based bioinformatics: Complement C1r/C1s, Uegf, Bmp1 (CUB), Pentraxin (PTX), Hormone Receptor (HormR), and GAIN[20,51,52]. However, a 150 aa region between PTX and HormR, could not be assigned to a known structural fold. Furin, a Golgi-localized protease, is reported to cleave human and mouse GPR126 in this region[51], although any effect on protein architecture is unclear because of the unspecified structure. In addition, alternative splicing occurs in Gpr126/GPR126, resulting in Gpr126/GPR126 isoforms that vary in their ECRs[51,53]. Alternative splicing of exon 6 was observed in human and zebrafish[30,51], producing isoforms that either include (S1 isoform, henceforth referred to as +ss) or exclude (S2 isoform, henceforth referred to as −ss) a 23 aa segment found within the unknown region between PTX and HormR. A genetic variant in GPR126 leading to decreased inclusion of exon 6 was recently found to be associated with adolescent idiopathic scoliosis[54]. Thus, determining the ECR structure, conformation, and other possible unexplored features will be instrumental in understanding Gpr126 function.

In this study, we determine the high-resolution crystal structure of the full-length ECR of zebrafish Gpr126, which reveals five domains, including a newly identified Sperm protein, Enterokinase and Agrin (SEA) domain, in which furin-mediated cleavage would occur in the human and mouse homologs. Intriguingly, the ECR is in an unexpected closed conformation that is reminiscent of the inactive closed conformation of the ECRs from EGFR and integrin families. This closed conformation is sustained by an alternatively spliced linker, while insertion of the alternatively spliced site gives rise to dynamic open-like ECR conformations and increases downstream signaling. A second feature that also mediates the closed conformation is a newly identified calcium-binding site at the tip of the ECR. Strikingly, zebrafish carrying point mutations at this site have both myelination defects and malformed ears, demonstrating the critical role of the ECR in Gpr126 function in vivo. These results altogether show that the ECR of Gpr126 has multifaceted roles in regulating receptor function, a feature that is likely true for other aGPCRs, and that will form the basis for further investigations in the efforts to drug aGPCRs.

## Results

**Structure of the full-length ECR of Gpr126.** To determine the structure of the ECR of Gpr126, the full-length ECR (−ss) from zebrafish Gpr126 (T39-S837) was expressed and purified from insect cells using the baculovirus expression system. Zebrafish Gpr126 (Fig. 1a) has high sequence identity (47%) to its human homolog but its ECR has a fewer number of N-linked glycosylation sites (15 predicted in zebrafish, 26 in human) and no furin-cleavage site (Supplementary Fig. 1A), and thus yields a more homogeneous sample (Supplementary Fig. 1B, C). Crystals of both native and selenomethionine (SeMet)-labeled zebrafish Gpr126 ECR (−ss) were obtained and diffracted to 2.4 Å (Supplementary Fig. 1D), and the structure was determined by SeMet single-wavelength anomalous diffraction (SAD) phasing (Supplementary Table 1).

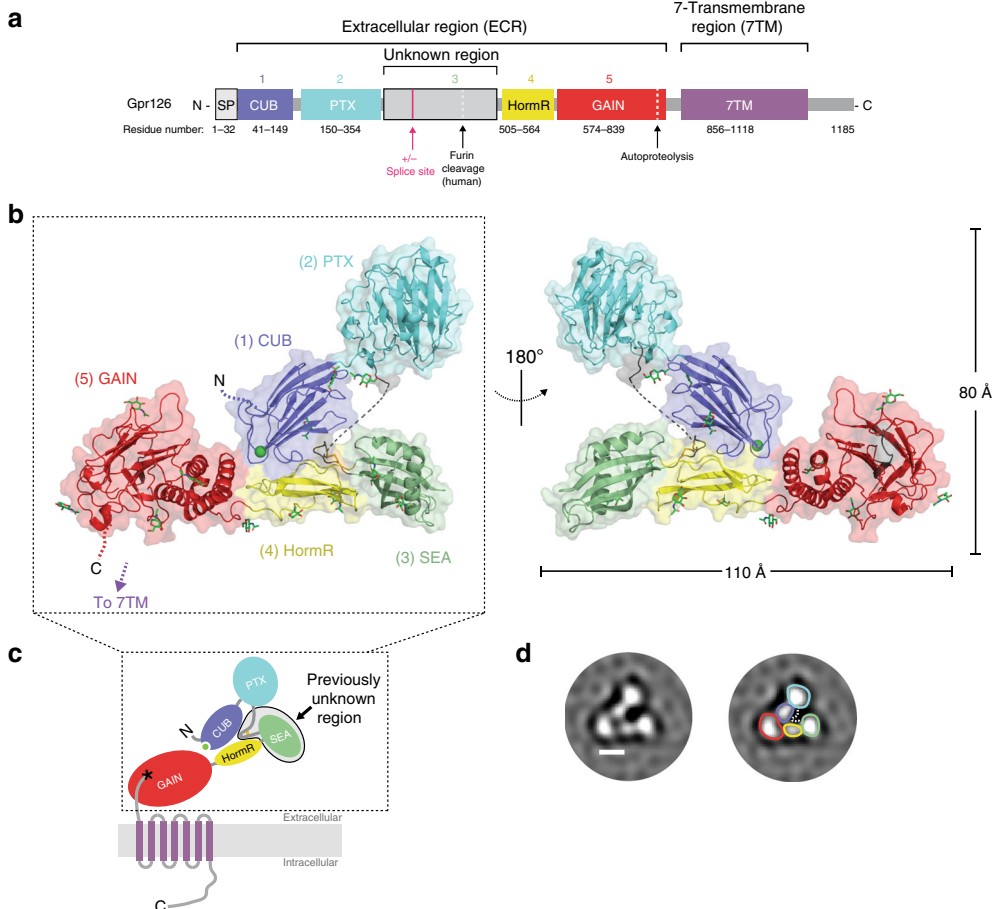

**Fig. 1 Crystal structure of the full extracellular region of Gpr126. a** Domain organization of Gpr126, indicating the ECR and 7TM regions. The unknown region includes a splice site. Cleavage sites (furin cleavage, autoproteolysis) are indicated by dashed lines. Domains are colored dark blue (CUB), cyan (PTX), gray (unknown region), yellow (HormR), red (GAIN), and purple (7TM). Domain boundaries are indicated below. SP indicates signal peptide. **b** Structure of the full ECR of (−ss) Gpr126. Domains are colored as in **a** except for the newly identified SEA domain (green). Domains are numbered (1–5) from N to C-terminus. Calcium ion in CUB domain is indicated as a green sphere. Dashed lines represent disordered residues. N-linked glycans are shown as green sticks. **c** Schematic of full-length Gpr126. The previously unknown region (SEA domain and linker region) is labeled. Autoproteolysis in GAIN domain is indicated by an asterisk and the last beta-strand of the GAIN domain is colored gray. **d** Representative negative-stain EM 2D class average of Gpr126 (−ss) ECR. Scale bar (white) represents 50 Å. Domains are assigned and colored according to color scheme noted above. The dashed line represents the linker region.

The structure, with overall dimensions of $110 \times 80 \times 35$ Å, revealed the presence of five domains (Fig. 1b, c), of which only four were identified previously. The N-terminal region of the protein is composed of the CUB domain followed very closely by the PTX domain. The 150 aa unknown region after the PTX domain was revealed to be a 22 aa linker that is partially disordered, the 23 aa alternatively spliced region (not present in crystal structure construct), and a structured domain which spans 105 aa and was identified as a SEA domain through the Dali server[55]. The Gpr126 SEA domain adopts a ferredoxin-like alpha/beta sandwich fold, common to SEA domains from other proteins. Interestingly, analysis of the structure as well as sequence alignments between zebrafish and human showed that furin cleavage in humans would occur in the SEA domain (Supplementary Fig. 1A). Finally, the SEA domain is followed by the HormR and GAIN domains, the latter of which is autoproteolyzed as expected (Supplementary Fig. 1E). The HormR and GAIN domain structures are similar to previously-solved HormR + GAIN domain structures from other aGPCRs[9,20], with the exception of the relative orientation between HormR and GAIN. There is a 90° rotation of the

HormR domain with respect to the GAIN domain (Supplementary Fig. 1F) in Gpr126 compared to previously-solved HormR + GAIN structures from rLphn1 and hBAI3[20]. In addition, Gpr126 was observed to have at least ten sites of glycosylation throughout all domains of the ECR except the PTX domain (Fig. 1b).

**Gpr126 (−ss) ECR adopts a closed conformation.** Unexpectedly, the structure revealed a compact, closed conformation where the most N-terminal CUB domain interacts with the more C-terminal HormR and GAIN domains (Fig. 1b). To ensure that this conformation is not a crystallization artifact, we utilized both negative-stain electron microscopy (EM) and small-angle X-ray scattering (SAXS) to confirm that the closed confirmation is observed for Gpr126 in solution. Negative-stain 2D class averages of Gpr126 ECR showed a V-shaped protein architecture (Fig. 1d). The individual domains in the 2D class averages were assigned according to size and are consistent with the closed architecture of the crystal structure. In addition, we measured the radius of gyration ($R_g$) of the ECR using SAXS to confirm that the closed conformation exists in solution. The observed $R_g$ ($41.1 \pm 0.1$ Å) is

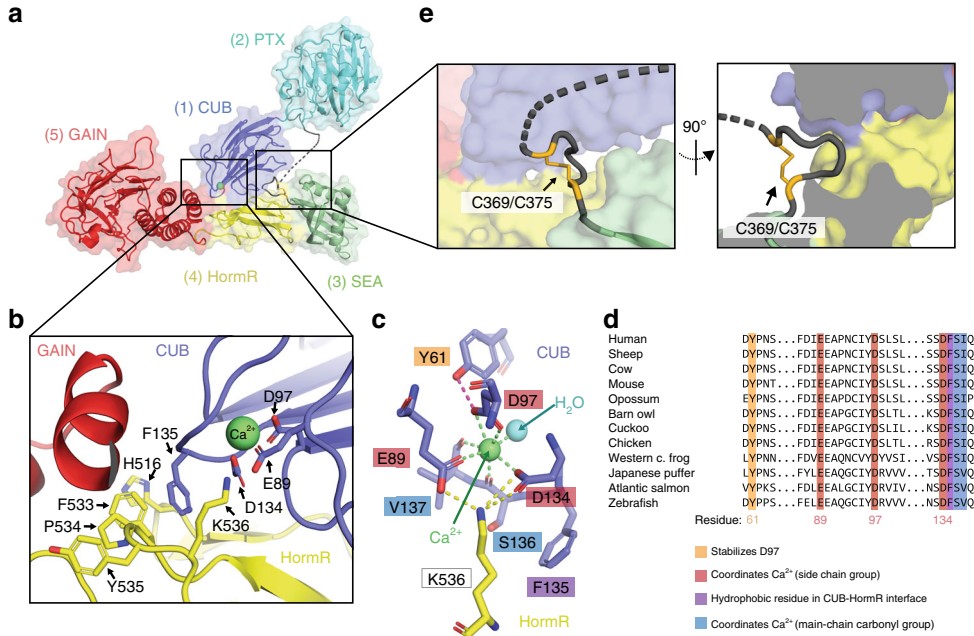

**Fig. 2 Closed conformation of Gpr126 is mediated by CUB-HormR-linker interactions. a** Structure of the full ECR of (−ss) Gpr126. **b** Close-up view of the CUB-HormR interface. Resides at the interface are shown as sticks. The calcium ion is shown as a bright green sphere. **c** Close-up view of the calcium-coordination site within CUB domain. The water molecule is shown as a blue sphere. The residues are shown as sticks. CUB residues are colored dark blue and HormR residue is colored yellow. Residue labels are colored according to their roles in CUB-HormR interaction: red (E89, D97, D134) represents calcium coordination by side-chain residue, blue (S136, V137) represents calcium coordination by main-chain carbonyl group, purple (F135) represents a hydrophobic residue in CUB-HormR interface, and orange (Y61) represents a residue that stabilizes calcium-coordinating residue D97. Calcium coordination is shown as bright green dashed lines. CUB-HormR interaction is shown as yellow dashed lines. The interaction between Y61 and D97 is shown as a magenta dashed line. **d** Sequence alignment of partial Gpr126 CUB domain from various species, highlighting important conserved residues: calcium-coordinating residues by side-chain group (red), calcium-coordinating residues by main-chain carbonyl (blue), a tyrosine residue that stabilizes a calcium-coordinating residue (orange), and a hydrophobic phenylalanine residue in the CUB-HormR interface (purple). **e** Close-up view of the disulfide-stabilized loop inserted between CUB and HormR domains. The disulfide bond is colored bright orange and is indicated by an arrow. The dashed line represents disordered residues in the linker region.

consistent with the calculated $R_g$ of the closed-conformation crystal structure model (42.6 Å) and inconsistent with that of an extended model of Gpr126 ECR in which the CUB domain points away, rather than toward, the center of the molecule ($R_g = 52.2$ Å) (Supplementary Fig. 1G). Taken together, these results: show that Gpr126 ECR is in a closed conformation in solution, demonstrate that this conformation is not an artifact of crystal-packing contacts, and suggest that this closed conformation may play an important role in Gpr126 function.

As the closed conformation of Gpr126 (−ss) ECR was shown to exist both in solution and in the crystal lattice, we next wanted to explore the interactions that contribute to this protein architecture. Close examination of the crystal structure revealed two interaction sites that mediate the closed conformation, the first of which is a direct interaction between domains that are at opposite ends of the ECR and the second is an indirect interaction formed between two domains through a loop that holds them together (Fig. 2a).

First, a direct interaction exists at the tip of the CUB domain (close to the N-terminus), which points inward towards the center of the molecule and lies in the interface between GAIN and HormR. Residues in the HormR domain (H516, F533, P534, Y535) interact with each other through pi-pi stacking (sandwich), promoting interaction with F135 on the CUB domain through additional (T-shaped) pi-pi stacking to stabilize the CUB-HormR interaction (Fig. 2b).

Surprisingly, examination of the 2Fo-Fc electron density map showed that there is density within the CUB domain at this

interface that does not belong to any amino acid residue (Supplementary Fig. 2A). This density is coordinated by the side-chain groups of E89, D97 (bidentate) and D134, main-chain carbonyl groups of S136 and V137, as well as a water molecule for a complex with coordination number 7 in a pentagonal bipyramid geometry (Fig. 2c). The geometry and distances between the density and the coordinating residues in Gpr126 are consistent with calcium coordination[56]. Several CUB domains from extracellular proteins are reported to coordinate calcium, including Gpr126[57], and some have been discovered to use this coordination to mediate ligand binding[57–61] (Supplementary Fig. 2B). For example, the C1s protein uses its CUB calcium-binding site to bind to ligand C1q and initiate the classical pathway of complement activation[61], and the Lujo virus recognizes a calcium-binding site on the CUB domain of the neurophilin-2 receptor in order to gain cell entry[59]. The calcium-coordinating residues are all conserved in the Gpr126 CUB domain (among GPR126 proteins from various species (Fig. 2d) as well as among calcium-binding CUB domains from other proteins (Supplementary Fig. 2C)), suggesting that the density is indeed calcium. Importantly, the calcium coordination aligns the coordinating residues E89 and D134 on the surface of the CUB domain such that they can interact with K536 on the HormR domain (Fig. 2c), contributing to the closed conformation.

In addition to the direct CUB-HormR interaction, a second interaction site is formed by a disulfide-stabilized loop, which provides a bridge between the CUB and HormR domains. Although 13 (C355-A367) of the 22 aa (C355-P376) in the linker

region are disordered in the structure, the rest were able to be resolved and they form a small loop stabilized by a disulfide bond between C369 and C375 (Fig. 2e). This loop is located directly N-terminal to the SEA domain and is inserted between the CUB and HormR domains, effectively bridging the two domains and likely contributing to the stabilization of the closed conformation. The cysteines that form the disulfide bond are conserved among all except four of the 94 species analyzed in this study (Supplementary Fig. 2D and Supplementary Data 1), suggesting that this disulfide bond plays an important role in Gpr126 function. The five residues (ASGLG) flanked by the cysteines are small and flexible, accommodating the formation of the disulfide loop as well as insertion into the small pocket between CUB and HormR.

**Alternative splicing modulates Gpr126 ECR conformation**. Gpr126 is alternatively spliced, producing several isoforms that may modulate protein function. Skipping of exon 6 results in deletion of 23 aa in zebrafish (28 aa in human) and is of particular interest because these amino acids reside in the previously unknown region of Gpr126 ECR. The 23 aa region is rich in serine/threonine residues (10 out of 23) and contains a predicted N-linked glycosylation site, which suggests that this region may be a highly O- and N-link glycosylated stalk. From analysis of the crystal structure (−ss isoform, in which the 23 aa are deleted), we determined that the splice site is directly between the regions encoding the disulfide-stabilized loop and the SEA domain (Fig. 3a). Because the disulfide-stabilized loop makes contacts that are important for the closed conformation of Gpr126 ECR (−ss) (Fig. 2e), we hypothesized that the (+ss) isoform would disrupt the closed conformation and have a different, more open conformation.

To test whether Gpr126 ECR (+ss) and (−ss) have different conformations, the two proteins were purified and analyzed using negative-stain EM. Single particles were classified into 2D class averages and the class averages were further categorized into groups to facilitate interpretation of different conformations. The class averages for the (−ss) isoform, categorized into five main orientations (Fig. 3b), were consistent with the closed conformation of the crystal structure (Fig. 1b). However, the class averages for the (+ss) isoform (Fig. 3c) showed a diverse population of ECR molecules, as they contain additional more open-like conformations (group vi, 21% of particles, Fig. 3d) as well as closed conformations that were observed in the (−ss) isoform (Fig. 3c, d). Furthermore, individual (+ss) particles showed the presence of open conformations (Fig. 3e), including a fully extended conformation, which could not be classified into a distinct class average during image processing. These results are consistent with our hypothesis that the (+ss) ECR conformation is different from that of (−ss) and suggest that the addition of 23 aa extends the linker in (+ss), likely disrupting the indirect and direct CUB-HormR interactions and preventing the stable closed conformation that is observed in (−ss) (Fig. 3f).

The negative-stain EM data are consistent with SAXS experiments showing that the $R_g$ of zebrafish Gpr126 ECR (+ss) is larger than that of (−ss) with a more dramatic change in $R_g$ observed between the human GPR126 isoforms (Supplementary Fig. 3A-F and Supplementary Table 2). Size-exclusion chromatography elution profiles for both zebrafish and human constructs also showed that (+ss) elutes earlier compared to (−ss), indicative of a larger size and different shape (Supplementary Fig. 3G, H).

**Alternative splicing modulates Gpr126 receptor signaling**. To determine whether the two isoforms also exhibit different levels of signaling, receptor activity was measured for both isoforms

using a G protein signaling assay. Human GPR126 has been shown previously to couple to and activate Gα<sub>s</sub>, leading to production of cAMP[42]. Therefore, we used a cAMP signaling assay in which HEK293 cells were co-transfected with a full-length zebrafish Gpr126 construct and a reporter luciferase that emits light upon binding to cAMP. Cell-surface expression levels of the constructs were quantified by flow cytometry analysis of cells stained by antibodies against N-terminal FLAG-tags (Fig. 4a and Supplementary Fig. 4A, B, C), and basal signaling results (Fig. 4b) were normalized to expression level (Fig. 4c).

Cells transfected with either (−ss) or (+ss) Gpr126 had higher cAMP levels compared to cells transfected with an empty vector (EV) (Fig. 4c), demonstrating that basal activity of Gpr126 can be detected in this assay. As a positive control, a synthetic peptide agonist that targets the 7TM activated zebrafish Gpr126 and human GPR126 signaling to a level consistent with similar, previously-published experiments on human GPR126[32] (Supplementary Fig. 4D) and did not activate signaling in EV-transfected cells.

However, the closed-conformation (−ss) Gpr126 signaled significantly less compared to the more dynamic (+ss) Gpr126 (Fig. 4c), and this result was consistent between both zebrafish and human constructs (Supplementary Fig. 4C and Fig. 4d–f). This suggests that the additional amino acids in the linker region of the ECR as a result of alternative splicing plays a role in modulating the activity of Gpr126 and that the ECR of Gpr126 is coupled to receptor signaling. Taken together with the negative-stain EM results, the (−ss) and (+ss) Gpr126 isoforms are distinct in terms of ECR conformation dynamics, as well as G protein signaling activity.

In addition, we mutated calcium-binding site residues D134A/F135A in the (−ss) isoform, which we predicted would disrupt the closed conformation. Using negative-stain EM, we observed open ECR conformations for this construct (Supplementary Fig. 4E-G), similar to the wild-type (+ss) isoform (Fig. 3c and Supplementary Fig. 4H). The calcium-binding site mutation did not increase or decrease the cAMP signaling for the (−ss) Gpr126 isoform, which suggests that the ECR conformation is not solely responsible for regulation of receptor signaling. However, the same mutation in the (+ss) isoform resulted in lower cAMP levels compared to wild-type (+ss) (Supplementary Fig. 4I). Cell-surface expression levels of these mutant Gpr126 constructs in HEK293 cells were similar or higher than wild-type constructs, excluding the possibility that lower signaling was due to improper protein folding or trafficking (Supplementary Fig. 4I). Altogether, these results might be explained by a complex, rather than a simple and straightforward, model of regulation for receptor signaling and suggest a possible functional role for the calcium-binding site.

**Calcium-binding site is critical for PNS myelination in vivo**. Functional sites on proteins are usually highly evolutionarily conserved. We used the ConSurf server[62] to perform surface conservation analyses on a diverse set of 94 Gpr126 protein sequences. The conservation score for each residue was mapped onto the Gpr126 ECR structure (Supplementary Fig. 5A), which revealed that the most conserved domain in the ECR is the CUB domain. Importantly, the calcium-binding site is absolutely the most highly conserved patch within the CUB domain and within the entire Gpr126 ECR (Fig. 5a). The calcium-binding site is universally conserved among all species analyzed, which suggests that the calcium-binding site has an essential role in Gpr126 function.

We next wanted to test whether the residues in the calcium-binding site are important for Gpr126 function in vivo. Gpr126 has previously been shown to regulate both PNS myelination and

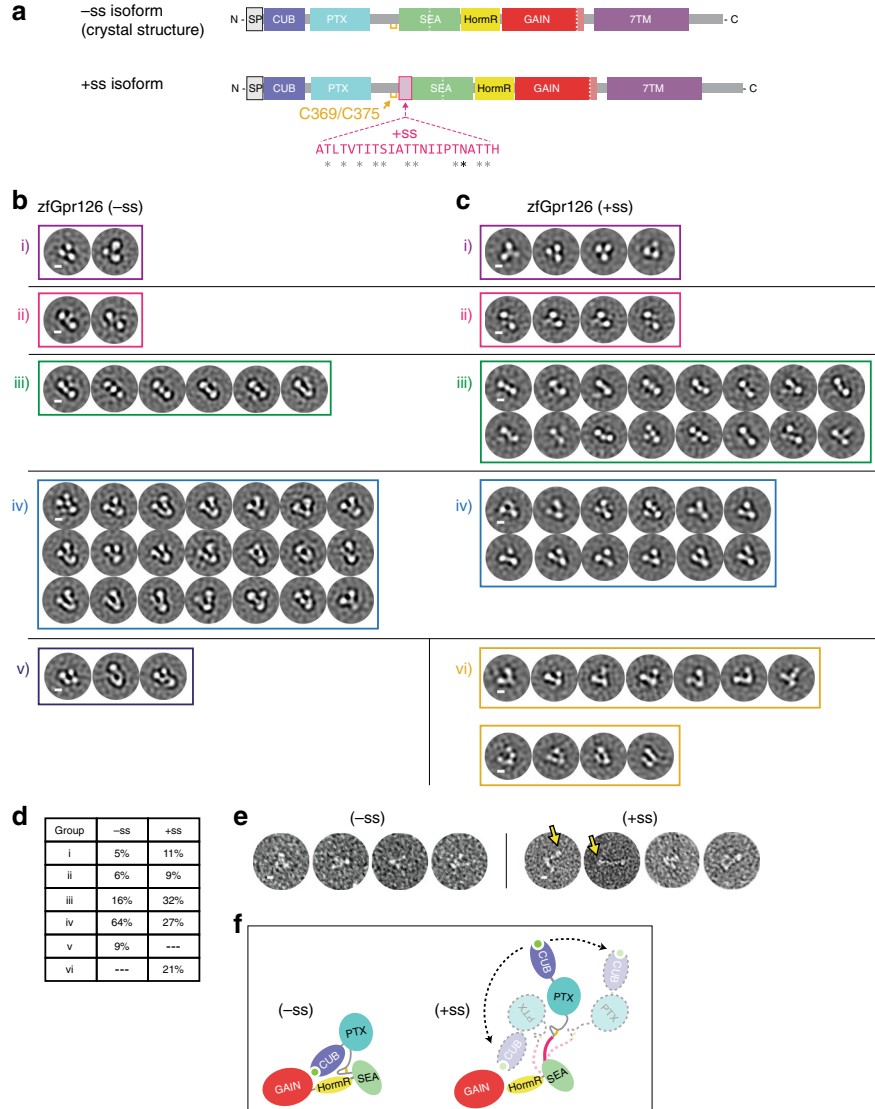

**Fig. 3 Alternative splice isoforms of Gpr126 modulate ECR conformation. a** Schematic diagram of Gpr126 splice isoforms generated by including (+ss) or excluding (−ss) exon 6. Residues encoded by exon 6 are colored magenta. Gray asterisks indicate potential O-linked glycosylation sites and the black asterisk indicates a predicted N-linked glycosylation site. The conserved disulfide bond in the linker is colored yellow. **b**, **c** Negative-stain EM 2D class averages for −ss (**b**) and +ss (**c**) ECR constructs. Class averages are categorized according to similar orientations: (i, ii, iii, iv, v, and vi). (i, ii, iii, iv) are observed in both −ss and +ss isoforms. (vi) represents open-like conformations (>50° angle) that are observed only in the +ss isoform. (v) represents unidentifiable miscellaneous views. Scale bars (white) represent 50 Å. **d** Quantification of percentage of particles per category for both isoforms.
**e** Representative individual particles for both isoforms. Yellow arrows point to particles which are not in a closed conformation. **f** ECR conformations based on negative-stain EM are depicted as cartoons. The splice site is shown in magenta. Black arrows with dashed lines indicate dynamic ECR conformation.

ear development in zebrafish through elevation of cAMP[35,36]. Zebrafish *gpr126* mutations that impair G protein signaling result in abolished myelination of the peripheral axons by SC and cause "puffy" ears[28,32,35,36,63]. GPR126 has been shown to have a role in heart development in mouse[37], supported by additional studies in zebrafish[28,30,63]. Gpr126 activity in zebrafish can be readily measured by analyzing the expression of *myelin basic protein (mbp)*, which encodes a major structural component of the myelin sheath and is essential for PNS myelination, and by assessing ear and heart morphologies of the fish. To determine whether the calcium-binding site is important for these functions, two amino acids in the site, D134 and F135, were targeted and mutated to alanines using CRISPR/Cas9-mediated homologous recombination. D134 directly coordinates the calcium ion and F135 is an adjacent hydrophobic residue which forms one arm of

the calcium-binding pocket (Figs. 2c, 5a). As a result, the mutant zebrafish, *gpr126^stl464*, harbor D134A and F135A mutations (Fig. 5b, Supplementary Fig. 5B). These mutations created a BstUI restriction enzyme site, which was used to genotype individual zebrafish (Fig. 5c). Expression of *gpr126* is unaffected in *gpr126^stl464* mutants (Supplementary Fig. 5C, D). Strikingly, compared to wild-type siblings, the *gpr126^stl464*-mutant zebrafish developed the puffy ears (Fig. 5d, e) that are indicative of a defect in Gpr126-mediated G protein signaling, though they do not appear to have heart defects (Supplementary Fig. 5E-T). In addition to the ear phenotype, mutant zebrafish did not express *mbp*, indicative of failed PNS myelination (Fig. 5f, g, Supplementary Fig. 5U, V). These results show that D134 and F135 in the calcium-binding pocket of Gpr126 are essential for ear and SC development in vivo.

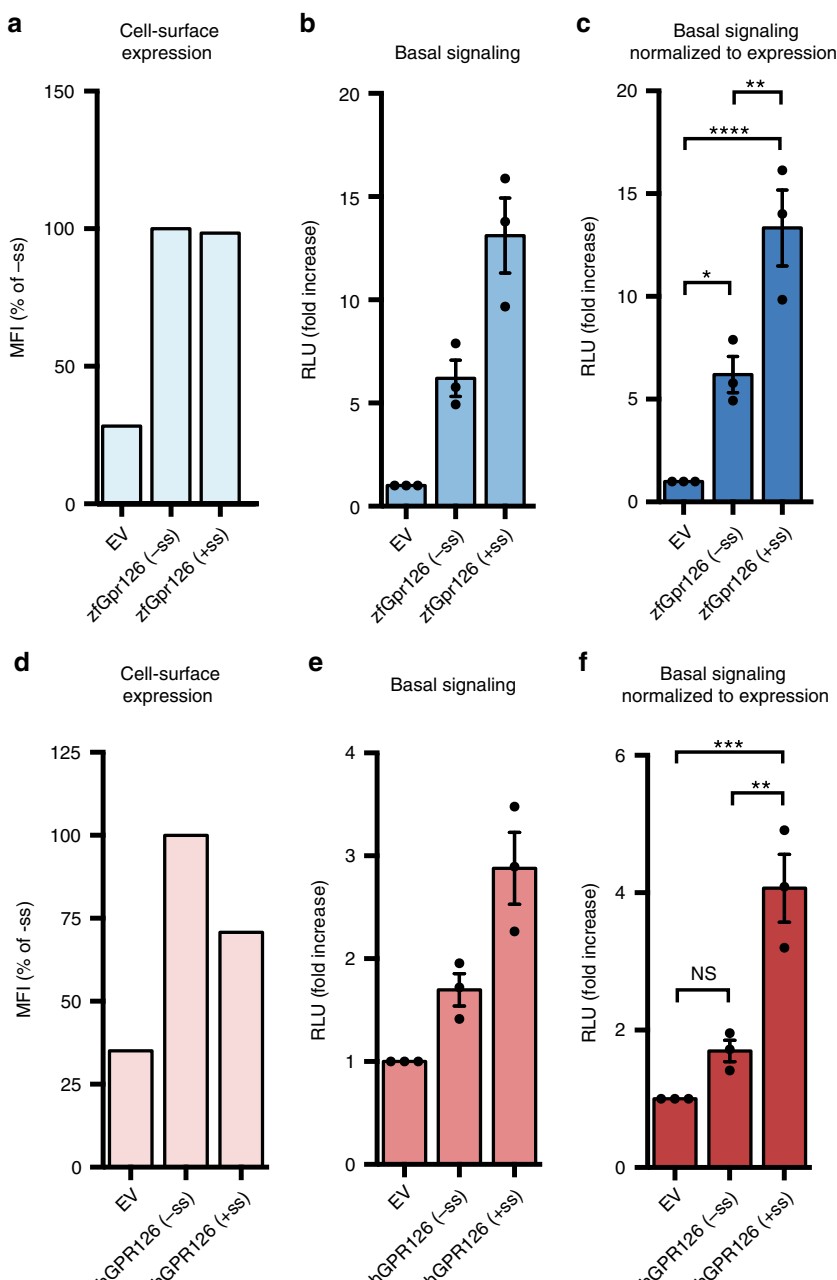

**Fig. 4 Alternative splice isoforms of Gpr126 modulate receptor signaling. a** Cell-surface expression levels for empty vector (EV), zebrafish Gpr126 splice isoforms, measured using flow cytometry to detect binding of anti-FLAG antibody to cells expressing FLAG-tagged Gpr126. The Gpr126 cell-surface expression levels are normalized to the control EV signal. Data are shown as MFI (mean fluorescence intensity). Error bars are not shown because expression levels are presented as median fluorescence intensities of 10,000 cells for each population of transfected cells, for a single flow cytometry experiment representative of at least three independent experiments. **b** Basal signaling measured by the cAMP signaling assay. Data are shown as fold increase over EV of RLU (relative luminescence units). **c** Basal cAMP signaling normalized to cell-surface expression. ns, $P > 0.05$; *$P \leq 0.05$; **$P \leq 0.01$; ***$P \leq 0.001$; ****$P \leq 0.0001$; by one-way ANOVA and Tukey's multiple comparisons test. Data in **b** and **c** are presented as mean ± SEM, $n = 3$, and are representative of at least three independent experiments. **d**–**f** Same as **a**–**c** but for human GPR126 splice isoforms. Source data are provided as a Source Data file.

**Identification of a proteolytic SEA domain in human GPR126.** As mentioned earlier, the previously unknown region in the Gpr126 ECR contains a structured domain, which we revealed to be a SEA domain (Fig. 6a). Gpr126 SEA superimposes well over known SEA domains from Mucin-1 and Notch-2[64,65], which are cleaved (via autoproteolysis and furin, respectively), both in the same loop between beta-strand 2 and beta-strand 3 (Fig. 6b, c). Although the GPR126 furin-cleavage site is conserved in many mammals and birds (Supplementary Data 1, Supplementary Fig. 1A), with a consensus sequence of (R/K)-X-K-R↓, it is not conserved in zebrafish Gpr126. Using sequence alignments (Fig. 6d) and homology modeling, we mapped the furin-cleavage site in human GPR126 (Fig. 6e, Supplementary Fig. 6A) to the same loop that is cleaved in Mucin-1 and Notch-2, suggesting that SEA domain cleavage plays similar roles in each of these proteins. Consistent with a previous study[51], R468A mutations

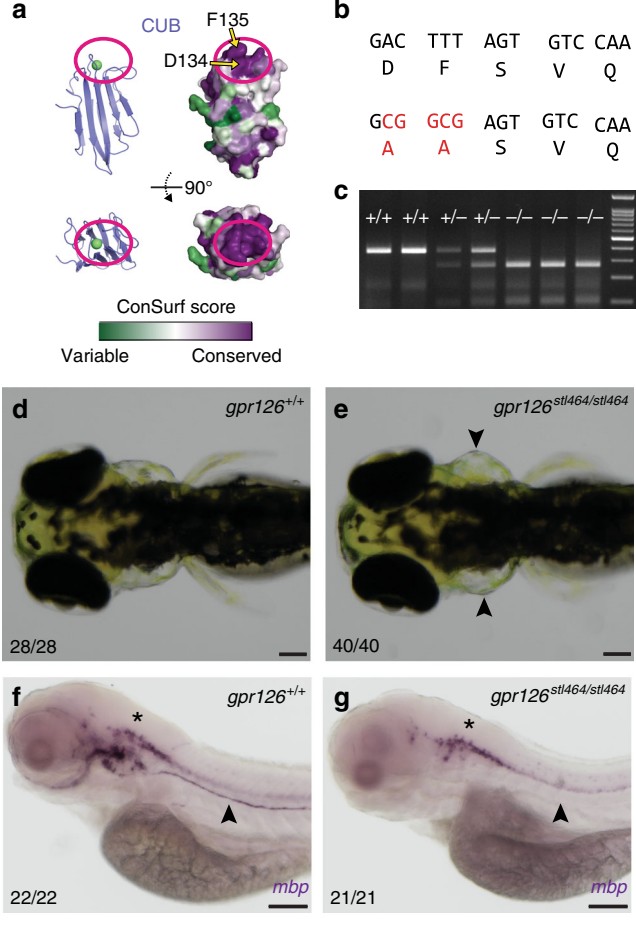

**Fig. 5 The calcium-binding site is required for Gpr126 function in vivo. a** Surface conservation analysis (green, variable; purple, conserved) of CUB domain. The calcium-binding site is circled in magenta. D134 and F135 are indicated by arrows. **b** D134 and F135 were both mutated to alanines through homologous recombination of a 150 bp ssODN containing a 5 bp mutation (red nucleotides). **c** Genotyping assay for the *gpr126*[stl464] lesion. The 5 bp mutation introduces a BstUI restriction enzyme binding site. **d** 4 dpf wild-type larva compared to **e** 4 dpf *gpr126*[stl464/stl464] larva with puffy ears (arrowheads). Scale bars (black) represent 100 μm. **f** 4 dpf wild-type larvae express *mbp* throughout the posterior lateral line nerve (PLLn, arrowhead), whereas **g** 4 dpf *gpr126*[stl464/stl464] larva lack *mbp* expression along the PLLn (arrowhead). Scale bars (black) represent 100 μm. Asterisks indicate CNS.

abolish furin cleavage in both human GPR126 (−ss) and (+ss) isoforms (Supplementary Fig. 6B, C). In addition, these mutant GPR126 constructs were transfected into HEK293 cells and were detected on the cell surface (Supplementary Fig. 6D), and therefore, the importance of furin cleavage is likely not primarily important for proper expression and trafficking.

To our knowledge, SEA and GAIN are the only known protein domains that are proteolyzed and remain associated even after proteolysis. In proteins like Mucins and Notch, the cleaved SEA domain remains intact[65,66] and shear forces likely unfold the domain and separate the protein into two fragments[67,68]. The Gpr126 SEA domain shows several noncovalent interdomain interactions, particularly between all four of the beta-strands that form a beta-sheet (Fig. 6f). The separation of the human GPR126 furin-cleaved SEA domain into two fragments does not readily occur immediately following cleavage as the cleaved protein resists separation when purified by size-exclusion chromatography (Supplementary Fig. 6B), similar to the aforementioned SEA domains as well as to GAIN domain autoproteolysis. Instead, the two fragments likely stay associated noncovalently until a disruptive event, such as ligand binding and mechanical force, unfolds the SEA domain and leads to separation or shedding of the region N-terminal to the furin-cleavage site (CUB, PTX, linker, half of SEA) and the C-terminal region (half of SEA, HormR, GAIN, 7TM).

## Discussion

aGPCRs make up the second largest family of GPCRs with 32 members in humans and are essential for numerous biological processes such as synapse formation, cortex development, neutrophil activation, angiogenesis, embryogenesis, and many more. Recent studies have shown that the ECRs of aGPCRs play important roles in these functions; however, the relative lack of information about the structures of ECRs and their mechanisms of activation hampers further studies toward drugging these receptors. Here we show that the large ECR of Gpr126, an aGPCR with critical functions in PNS myelination, ear development, and heart development, adopts an unexpected closed conformation where the most N-terminal CUB domain interacts with the more C-terminal HormR domain. The structure of the Gpr126 ECR revealed that the closed conformation is mediated through a calcium-binding site as well as a disulfide-stabilized loop. Interestingly, the residues involved in these intramolecular interactions are highly conserved among Gpr126 sequences, including that of zebrafish, raising questions about their role in Gpr126 function.

Alternative splicing is a mechanism to increase the functional diversity of metazoan genomes and has been repeatedly demonstrated to play a role in the regulation of brain function. For example, alternative splicing contributes to the functional diversification of DSCAMs, protocadherins, calcium channels, neurexins, and neuroligins[69–72]. It is also proposed that alternative splicing may cause a large conformational change in the ECR of the synaptic protein teneurin, since alternative splicing allows the protein to act as a switch in regulating ligand binding despite the ligand-binding site being distant from the seven aa alternatively spliced site[73]. Since *gpr126* is alternatively spliced in the region encoding the ECR, we examined the functional differences between isoforms. Our negative-stain EM and SAXS results suggest that alternative splicing between the regions encoding the PTX and SEA domains in *gpr126* perturbs the closed conformation and generates a population of ECR conformations that range from closed to extended (Fig. 7a). Several of the inserted residues resulting from alternative splicing are predicted to be sites of glycosylation. These glycosylation sites as well as the state of the other glycosylation sites may contribute to the change in ECR conformation. Our signaling assay results also show that alternative splicing leads to changes in basal receptor activity, which suggests that the architecture and conformation of aGPCR ECRs play more important roles in their functions than previously thought. However, the signaling assay results showing that the change of Gpr126 ECR conformation is not solely responsible for changes in signaling may be confusing and contradictory. Rather, a more complex model that combines changes in ECR conformation with exposure of potential functional sites due to these changes may be key for alternative splicing-mediated regulation.

Importantly, we identified the calcium-binding site in Gpr126 as a potential functional site. Our in vivo results showed that zebrafish carrying two point-mutations in the calcium-binding site have defective SC and ear development, suggesting that the calcium-binding site is essential for the in vivo functions of Gpr126 (Fig. 7b). Since a subset of CUB domains from other

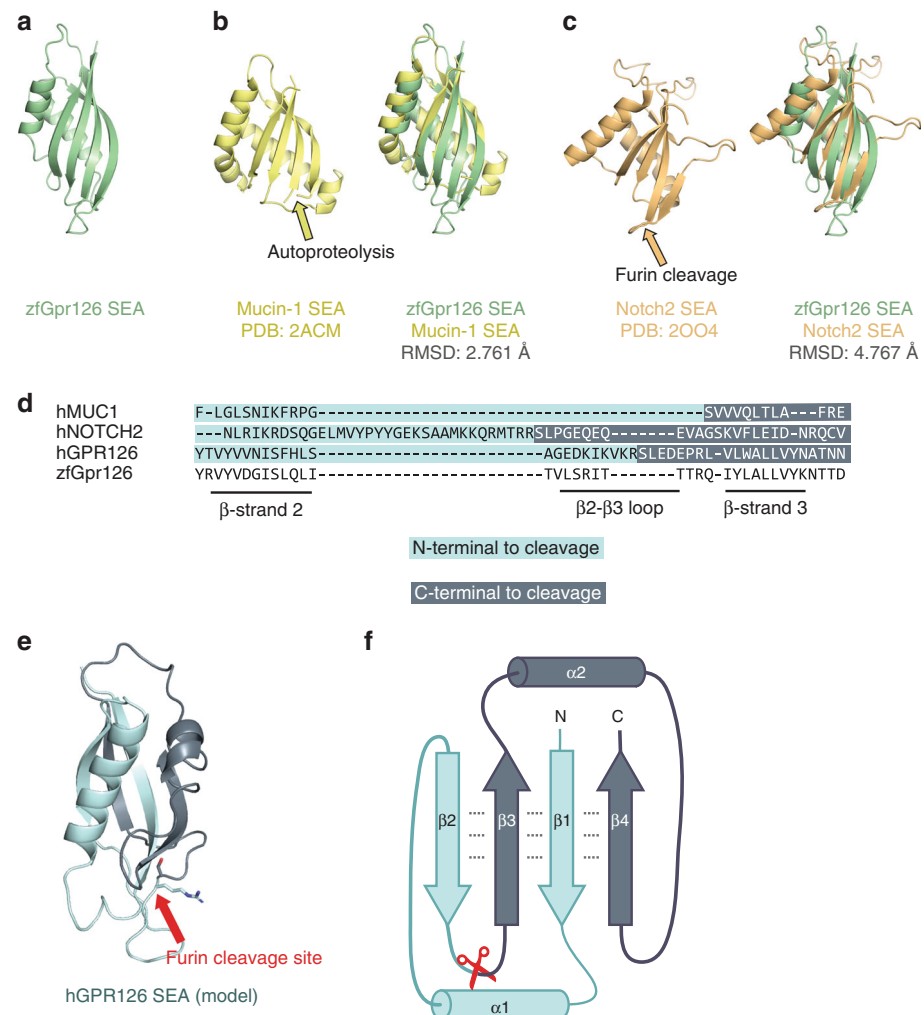

**Fig. 6 Identification of a proteolytic SEA domain in human GPR126. a** Crystal structure of the SEA domain from zebrafish Gpr126. **b** (left) NMR structure of Mucin-1 SEA domain (PDB: 2ACM) and (right) Gpr126 SEA domain superimposed over Mucin-1 SEA domain. The loop containing the autoproteolysis site in Mucin-1 is indicated by a yellow arrow. The root-mean-square deviation (RMSD) of atoms between overlaid structures is 2.761 Å. **c** (left) Crystal structure of the Notch2 SEA domain (PDB: 2OO4) and (right) Gpr126 SEA domain superimposed over Notch2 SEA domain. The loop containing the furin-cleavage site (deleted in crystal structure construct) is indicated by an orange arrow. The RMSD of atoms between overlaid structures is 4.767 Å. **d** Sequence alignment of partial SEA domain from human Mucin-1, human Notch2, human GPR126, and zebrafish Gpr126. **e** Homology model of human GPR126 SEA model generated using SWISSMODEL. The arrow points to modelled furin-cleavage site. **f** Protein topology map of SEA domain. Furin-cleavage site is indicated by red scissors. Residues N-terminal to cleavage site are dark blue and residues C-terminal to cleavage site are light blue. Dashed lines represent backbone hydrogen bonds between beta sheets.

proteins coordinate calcium in order to mediate ligand-binding[57], one possibility for the critical function of the calcium-binding site in Gpr126 may be to act as a ligand-binding site as well, although future experiments will need to be performed to validate this hypothesis.

The structure also revealed the presence of a SEA domain. In human and other species, a furin-cleavage site is mapped to this domain but this cleavage site is not conserved in zebrafish. Therefore, the function of the furin-cleavage may play a role in GPR126 that is not conserved in zebrafish. Cleaved SEA domains from other proteins have been shown to stay intact until a force is applied and pulls apart the fragments[67,68]. Similarly, GPR126 may regulate its activity by furin-dependent shedding in addition to the established GAIN-autoproteolysis-dependent shedding. Moreover, the released extracellular fragments may act as diffusible ligands and bind to other cell-surface receptors, but further studies need to be done to test this model. Other aGPCRs that

have SEA domains in their ECRs include ADGRF1/GPR110 and ADGRF5/GPR116[74,75]. Although these SEA domains are not cleaved by furin, they do contain the GSVVV (or GSIVA) motif that leads to autoproteolytic cleavage in the same loop (between beta-strand 2 and beta-strand 3) that is cleaved by furin in GPR126. Therefore, SEA domain cleavage, whether by autoproteolysis or by furin, is a common feature in several aGPCRs and may have similar roles in regulating receptor function.

Taken together, our results suggest that Gpr126 is a complex protein that makes use of its many domains to regulate its function. In addition to the autoproteolysis-dependent activation mechanism (Supplementary Fig. 7A), Gpr126 uses other mechanisms to regulate its function including modulation of the ECR conformation. In the closed conformation, Gpr126 signals less compared to when the ECR is in a more dynamic, open conformation, which may be regulated by alternative splicing (Fig. 7a). Alternative splicing which deletes the CUB domain[30]

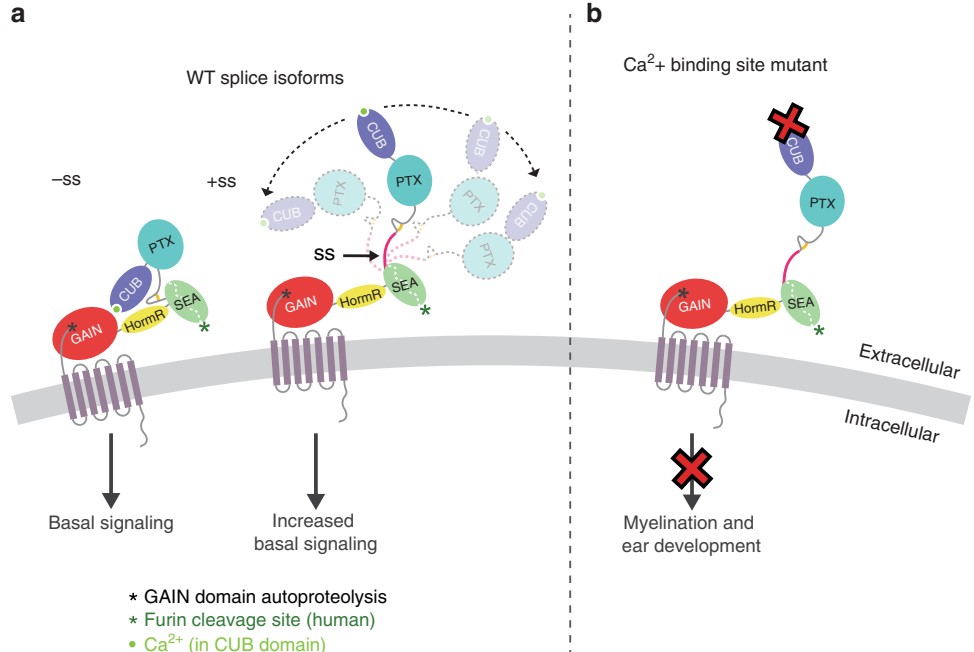

**Fig. 7 Model for ECR-dependent functions of Gpr126.** The model depicts how Gpr126/GPR126 function is regulated by its ECR. **a** Alternative splicing acts as a molecular switch to adopt different ECR conformations and have different basal levels of signaling. Gpr126 ECR that lacks the splice insert adopts a closed conformation and has basal activity, whereas Gpr126 ECR that includes the splice insert (magenta) is more dynamic and open-like, and has enhanced basal activity. Autoproteolysis in GAIN domain is indicated by a black asterisk. The calcium ion is shown as a bright green circle. **b** Mutation of conserved residues within the calcium-binding site leads to defects in both myelination and ear development in vivo. Human GPR126 function may also be regulated by furin cleavage, indicated by a green asterisk.

may also regulate receptor function (Supplementary Fig. 7B). Mutation of the calcium-binding site leads to signaling defects in vitro and to ear and PNS defects in vivo (Fig. 7b)). In addition, furin cleavage may allow GPR126 another mode of activation that is common to other receptors and adhesion GPCRs.

The Gpr126 closed conformation and hidden calcium-binding site is conceptually similar to EGFR. EGFR is in a closed, compact inactive conformation until ligand binding leads to a conformational change that extends the protein and reveals a hidden functional site that is important for its activation[10,76]. Because this mechanism is key for drugging EGFR, the conceptual similarity provides an opportunity to also drug Gpr126. Drugs that alter the ECR conformation of Gpr126 or block functional sites, such as the calcium-binding site, may be useful for treating Gpr126-associated diseases. The ECRs of other aGPCRs are major players in mediating receptor functions as well. For example, using its ECR, ADGRA2/GPR124 regulates isoform-specific Wnt signaling[77–80], the *C. elegans* ADGRL1/LAT-1 controls cell division planes during embryogenesis, and ADGRB1/BAI1 and ADGRL3/Lphn3 mediate synapse formation through interaction with other cell-surface proteins[81–84]. Thus, the ECRs of other aGPCR family members are also promising drug targets to treat numerous diseases once mechanistic details about their regulatory functions are understood.

## Methods

**Cloning and purification of Gpr126/GPR126 from insect cells.** The ECRs (residues T39-S837) of zebrafish Gpr126 and ECRs (residues C38-A853) of human GPR126, along with C-terminal 8XHis-tags, were cloned into the pAcGP67a vector. The following primers were used for amplification of zebrafish Gpr126: F: 5'-CGCATTCTGCCTTTGCGGCGAGCACCAGCTGCAATGTGGT-3' and R: 5'-GAATTCTAGAAGGTACCCGGTTAGTGGTGGTGATGGTGATGATGATGA GCTCTAGAGACATCCATTAGGATGCC-3'. The following primers were used for amplification of human GPR126: F: 5'-TCTGCCTTTGCGGCGAGCACCA

GCTGTAGGGTTGTCCTGAGCAACCCG-3' and R: 5'-GTGGTGGTGATGGTG ATGATGATGGGACCGTGGCAGATCCATAAGCAC-3'. Human GPR126 R468A mutant ECRs were generated using the QuikChange method (Agilent) with primers: F: 5'-GACAAGATTAAGGTGAAGGCGTCTTTGGAGGACGAGCC-3' and R: 5'-GGCTCGTCCTCCAAAGACGCCTTCACCTTAATCTTGTC-3'.

All proteins were expressed using the baculovirus method. Sf9 cells (Thermo Fisher, 12659017) were co-transfected with the constructed plasmid and linearized baculovirus DNA (Expression Systems, 91-002) using Cellfectin II (Thermo Fisher, 10362100). Baculovirus was amplified in Sf9 cells in SF-900 III medium supplemented with 10% FBS (Sigma–Aldrich, F0926). High Five cells (Thermo Fisher, B85502) at a density of $2.0 \times 10^6$ cells ml$^{-1}$ in Insect-XPRESS medium (Lonza, 12-730Q) were infected with high-titer baculovirus and incubated for 72 h at 27 °C. All subsequent steps are conducted at 25 °C. The cells were pelleted at $900 \times g$ for 15 min and the conditioned medium containing the secreted glycosylated proteins were collected. To the medium were added final concentrations of 50 mM Tris pH 8, 5 mM CaCl$_2$, and 1 mM NiCl$_2$. The mixture was stirred for 30 min and then centrifuged at $8000 \times g$ for 30 min. The clarified supernatant was incubated with nickel-nitrilotriacetic agarose resin (Qiagen, 30250) for 3 h. The resin was collected with a glass Buchner funnel and washed with HBS buffer (10 mM HEPES pH 7.2, 150 mM NaCl) containing 20 mM imidazole. Purified protein was eluted with HBS buffer containing 200 mM imidazole and run on size-exclusion chromatography (Superdex 200 10/300 GL; GE Healthcare) in HBS buffer.

Selenomethione-labeled Gpr126 (−ss) ECR was expressed as previously described[85]. Briefly, High Five cells in Insect-XPRESS medium were adapted to ESF921 medium (Expression Systems, 96-001-01). The cells were subsequently centrifuged at $100 \times g$ for 15 min and resuspended in ESF921 methionine-free medium (Expression Systems, 96-200). The cells were expanded in the same medium and then infected at a density of $2.0 \times 10^6$ cells ml$^{-1}$ with high-titer baculovirus. At 10 h post-infection, 100 mg Seleno-L-methionine (Sigma–Aldrich, S3132) was added to each liter of cell culture. At 36 h post-infection, another 150 mg Seleno-L-methionine was added to each liter of cell culture. The cells were harvested 72 h post-infection and the purification process was the same as described above.

**X-ray crystallography.** Purified Gpr126 (−ss) ECR (both native and SeMet-labeled) was crystallized at 3 mg mL$^{-1}$ in 50 mM potassium dihydrogen phosphate, 20% (w/v) PEG 8000. Both native and SeMet-labeled datasets were collected to 2.4 Å at the Advanced Photon Source at Argonne National Laboratory (beamline 23-ID-D). The datasets were processed with HKL2000 and an initial model was

determined by SAD phasing using Crank2 in CCP4. Refinement was performed with both REFMAC5 (CCP4) and phenix.refine (PHENIX).

**Negative-stain electron microscopy.** Uranyl formate (0.75%) solution was freshly prepared by adding 5 mL boiling water to 37.5 mg uranyl formate (Electron Microscopy sciences, 22450). After stirring for 5 min in the dark, 10 μL 5 M NaOH was added and stirred for an additional 5 min. The solution was syringe filtered (Millipore, SLGV033RS) and stored in the dark. Purified Gpr126 (−ss), (+ss), and (−ss) D134A/F135A ECR constructs were diluted to ~5 ug mL$^{-1}$ and applied to glow-discharged EM grids (Electron Microscopy Sciences, CF400-Cu,) using a conventional negative-stain protocol[86]. To the grid was applied 2 μL diluted protein for 30 sec. The protein was blotted off with filter paper (Sigma–Aldrich, WHA1001110), and then the grid was touched to a 25 μL drop of distilled, filtered water. The water was blotted off, and the grid was touched to a second 25 μL drop of water and blotted off. The grid was then touched to a 25 μL drop of 0.75% uranyl formate for 30 sec and blotted off. The grid was air-dried for 30 sec. The sample was imaged on a Tecnai G2 F30 operated at 300 kV. Gpr126 −ss (6565 particles), +ss (2529 particles), and −ss D134A/F135A (3916 particles) were processed using EMAN2[87].

**Small-angle X-ray scattering.** SAXS measurements were performed at the Advanced Photon Source at Argonne National Laboratory (beamline 18-ID) with an in-line SEC columns (Superdex 200 or Biorad EnRich 5–650 10–300) equilibrated with 20 mM HEPES, pH 7.4, and 150 mM NaCl. Data were analyzed using autorg and datgnom using the commands "autorg –sminrg 0.55 –smaxrg 1.1" and "datgnom '1'.dat -r '2' – skip '3' -o '1'.out," respectively, where '1' is the file name, '2' is the $R_g$ determined by autorg, and '3' is the number of points removed at low q as determined from autorg. SAXS curves of molecular models were generated with Crysol version 2.83[88].

**cAMP signaling assay.** Full-length wild-type and mutant Gpr126/GPR126 constructs were cloned into pCMV5. All constructs include N-terminal FLAG-tags for measuring cell-surface expression levels. The following primers were used for amplification of zebrafish Gpr126: F: 5'-GCTGACTACAAAGACGATGACGACA AGCTTTGCAATGTGGTGCTCACCGACTCCCAGGGC-3' and R: 5'-CCTGGCC AGGCCTCTGGTCCATGAGGCCCCTTATTGCAGGGTACTATCTGCATTACT GTG-3'. The following primers were used for amplification of human GPR126: F: 5'-GCTGACTACAAAGACGATGACGACAAGCTTTGCGCAAACTGTAGGGT TGTCCTGA-3' and R: 5'-CAGGCCTCTGGTCCATGAGGCCCCTCAACAGGG GCCAGTTTTCACCAG-3'. Zebrafish Gpr126 D134A/F135A mutant constructs were generated using the QuikChange method (Agilent) with primers: F: 5'-GAT GGAGGTTTTCTTTAACTCCGCCGCTAGTGTCCAAAAGAAAGGCTTCC-3' and R: 5'-GGAAGCCTTTCTTTTGGACACTAGCGGCGGAGTTAAAGAAAA CCTCCATC-3'.

HEK293 cells (ATCC CRL-1573) were seeded in 6-well plates with Dulbecco's Modified Eagle Medium (DMEM; Gibco, 11965092) supplemented with 10% FBS (Sigma–Aldrich, F0926). At 60–70% confluency, the cells were co-transfected with 0.35 μg Gpr126 DNA, 0.35 μg GloSensor reporter plasmid (Promega, E2301), and 2.8 μL transfection reagent Fugene 6 (Promega, PRE2693). After a 24-h incubation, the transfected cells were detached and seeded (50,000 cells per well) in a white 96-well assay plate. Following another 24-h incubation, the DMEM was replaced with 100 μL Opti-MEM (Gibco, 31985079) and incubated for 30 min. To each well was then added 1 μL GloSensor substrate and 11 μL FBS. Basal-level luminescence measurements were taken after 30 min to allow for equilibration. For activation assays, the cells were then treated with either 1 mM p14 synthetic peptide (GenScript, N-THFGVLMDLPRSASEKEK-Biotin-C) or vehicle DMSO for 15 min. Measurements were taken with a Synergy HTX BioTeck plate reader at 25 °C.

**Flow cytometry to measure cell-surface expression of Gpr126.** HEK293 cells were transfected as described above and incubated for 24 h. The cells were then detached and seeded in a 24-well plate. Following another 24-h incubation, the cells were detached with citric saline and washed with phosphate-buffered saline (PBS). The cells were then washed twice with PBS supplemented with 0.1% BSA (Sigma–Aldrich, A3803). The cells were incubated with mouse anti-FLAG primary antibody (1:1000 dilution in PBS + 0.1% BSA; Sigma–Aldrich, F3165) at room temperature for 30 min and washed twice with PBS + 0.1% BSA. The cells were then incubated with donkey anti-mouse Alexa Fluor 488 secondary antibody (1:500 dilution in PBS + 0.1% BSA; Invitrogen, A21202) at room temperature for 30 min and washed twice with PBS + 0.1% BSA. Stained cells were resuspended in PBS + 0.1% BSA and were analyzed with a BD Accuri C6 flow cytometer.

**Zebrafish rearing.** Zebrafish were maintained in the Washington University Zebrafish Consortium Facility (http://zebrafish.wustl.edu), and the following experiments were performed according to Washington University animal protocols. All zebrafish experiments were performed in compliance with institutional ethical regulations for animal testing and research at Washington University and Oregon Health and Science University (OHSU). Experiments were approved by the Animal Care and Use Committee of Washington University School of Medicine (St. Louis, MO) and the Institutional Care and Use Committee of OHSU (Portland,

OR). The gpr126$^{stl464}$ zebrafish were generated within the wild-type AB* background. All crosses were either set up as pairs or harems and embryos were raised at 28.5 °C in egg water (5 mM NaCl, 0.17 mM KCl, 0.33 mM CaCl$_2$, 0.33 mM MgSO$_4$). Larvae were staged at days post fertilization (dpf). gpr126$^{stl464}$ larvae can be identified at 4 dpf by a puffy ear phenotype.

**Genotyping.** To identify carriers of the gpr126$^{stl464}$ allele, the following primers were used to amplify the 381 base pair (bp) locus of interest: F: 5'-GTTGTCG TCAAGACCGGCAC-3' and R: 5'- TCCACCTCCCAGCTACAATTCC-3'. After amplification by PCR, the product was digested with either DrdI (NEB) at 37 °C or BstUI (NEB) at 60 °C, and then run on a 3% agarose gel. The mutation both disrupts a DrdI binding site and introduces a BstUI binding site. DrdI cleaves wild-type PCR product into 275 and 105 bp products, and the mutant product is 380 bp. BstUI cleaves mutant PCR product into 274 and 106 bp products, and the wild-type product is 380 bp. We recommend using BstUI for genotyping. Any larvae identified with the puffy ear phenotype were always genotyped as gpr126$^{stl464}$ homozygous mutant ($n = 20/20$).

**Guide RNA synthesis.** Potential gRNA templates were generated by CHOPCHOP (http://chopchop.cbu.uib.no/). The chosen forward and reverse oligonucleotides, 20 bps upstream of the PAM sequence, were ordered with additional nucleotides added to the 5' end to permit cloning into the pDR274 vector[89]. The oligonucleotide forward sequence used was: 5' - tag gAC TTT AGT GTC CAA AAG AA - 3' and oligonucleotide reverse sequence used was: 5'- aaa cTT CTT TTG GAC ACT AAA GT – 3'. 2 uM of each oligonucleotide was mixed in annealing buffer (10 mM Tris, pH 8, 50 mM NaCl, 1 mM EDTA) and incubated at 90 °C for 5 min, then cooled to 25 °C over a 45 min time interval. The pDR274 vector was linearized with BsaI and oligonucleotides were ligated into the vector with T4 ligase (NEB) for 10 min at room temperature. The ligation reaction was transformed into competent cells and then plated on kanamycin LB plates. Selected colonies were grown, miniprepped (Zyppy Plasmid Kits, Zymo Research), and Sanger sequenced. The gRNA DNA sequence was then PCR amplified from 50 ng μL$^{-1}$ of the plasmid with Phusion (NEB) and the following primers: F: 5'-GTTGGAACCTCTTACGT GCC-3' and R: 5'-AAAAGCACCGACTCGGTG-3'. The PCR product was digested with DpnI at 37 °C for 1 h, heat inactivated at 80 °C for 20 min, and then purified with a Qiagen PCR Purification column. RNA was synthesized with a MEGAscript T7 Transcription Kit (Ambion).

**Design of ssODN and microinjections.** One-cell stage wild-type embryos were injected with either 2 or 3 nl of a solution containing ~132 ng μL$^{-1}$ gRNA, ~148 ng μL$^{-1}$ of Cas9 mRNA (obtained from the Hope Center Transgenic Core at Washington University in St. Louis), and 60 ng μL$^{-1}$ of the ssODN. The 150 bp ssODN was ordered from IDT and contained a 5 bp mutation (uppercase): 5'-atcat aaacataccttgcttgtaactgatatggaagcctttcttttggacactCGCCGcggagttaaagaaaacctccatcac atttccagtggagttgag-3'. Please note that an extra C (bolded), beginning after exon 3, is present in the ssODN that is not present in the gpr126 reference sequence. The extra nucleotide was not integrated into the stl464 mutants. At 1 dpf, embryos were genotyped for disruption of the wild-type DrdI binding site and screened for the characteristic gpr126 puffy ear mutant phenotype. Mutations that were successfully transmitted to the F1 offspring were screened for by restriction enzyme digest analysis. Mutant bands were gel extracted (Qiagen Gel Extraction Kit) and Sanger sequenced to identify the incorporation of the ssODN containing the mutation of interest.

**Whole mount in situ hybridization.** 1 dpf larvae were treated with 0.003% phenylthiourea to inhibit pigmentation until fixation in 4% paraformaldehyde at 4 dpf. After fixation, larvae pooled in microcentrifuge tubes were dehydrated in methanol (5 by 5 min washes while nutating) and then stored at −20 °C. To begin in situ hybridization, larvae were re-hydrated with 50, 70, and 100% 0.2% PBS-Tween: methanol (5 min washes on a nutator). PBS-Tween washes were then continued (4 by 5 min washes while nutating). Larvae were then treated with 1:900 Proteinase-K (20 mg/mL) in PBS-Tween for 45 min at room temperature (not on a nutator). After Proteinase-K treatment, two quick PBS-Tween washes were performed, and then larvae were post-fixed in 4% paraformaldehyde for 20 min while nutating. Larvae were then washed with PBS-Tween (5 by 5 min washes on a nutator) and then incubated in hybridization buffer (Hyb(+)) at 65 °C for at least 1 h in a dry heat block. The riboprobe of interest in Hyb(+) was then incubated with the larvae overnight at 65 °C in a dry heat block. The following day, larvae were washed in 100% Hyb(+), 75% Hyb(+): 25% 2X SSCTween, 50% Hyb(+): 50% 2X SSCTween, 25% Hyb(+): 75% 2X SSCTween, all preheated to 65 °C, in a dry heat block. Larvae were then washed with 2X SSCTween (two washes, 30 min each) and 0.2X SSCTween (two washes, 30 min each) at 65 °C in a dry heat block. Larvae were then washed with MAB-TritonX-100 for 10 min at room temperature while nutating and then blocked in blocking solution (2% blocking reagent in MAB + 0.2% TritonX-100 + 10% sheep serum) for at least 1 h at room temperature on a nutator. Larvae were then treated with anti-DIG AP Fab fragments (1:2000 – Roche 11093274910) in blocking solution overnight at 4 °C on a nutator. The following day, larvae were washed with MAB-TritonX-100 (6 by 30 min washes on a nutator) at room temperature. After a 10 min wash in alkaline phosphatase/NTMT (AP)

buffer, larvae were moved to a 24-well plate, covered in aluminum foil to prevent light exposure, and incubated with NBT ($2.2\,\mu L\,mL^{-1}$) + BCIP ($1.6\,\mu L\,mL^{-1}$) in AP buffer until the reaction completed. After development of the probe, larvae were washed with three quick PBS-Tween washes and then passed through 30, 50, and 70% glycerol washes. A complete protocol with detailed notes is available[90]. The previously characterized riboprobes utilized in this study were *mbp* (GenBank: AY860977.1)[91], and *gpr126*, originally synthesized with 5'- ggaattcgtgatggagctggt-gaacatagc-3' and 5'-agtgtcgactcacttctcatctatcaactcagcagc-3' primers[36]. For *mbp*, larvae were scored for either presence or absence of signal expression along the PLLn.

**Reporting summary**. Further information on research design is available in the Nature Research Reporting Summary linked to this article.

## Data availability

The accession number for the coordinates and diffraction data for the Gpr126 (−ss) ECR crystal structure reported in this paper is PDB: 6V55. The SASBDB IDs for the SAXS experimental data are: SASDFT9, SASDFU9, SDSDFV9, SASDFW9, SASDFX9. The source data underlying Fig. 4b, c, e, f and Supplementary Figures 1B, 4C, H, and 6B, C are provided as a Source Data file. All other data are available from the corresponding author on reasonable request

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

## Acknowledgements

We thank Engin Özkan for assistance with crystal structure determination and for insightful discussions, the He lab for the use of their luminescence plate reader, and other members of the Araç lab for helpful discussions. We also thank the staff at the University of Chicago Advance Electron Microscopy Facility, the Advanced Photon Source (APS) at Argonne National Labs (ANL), the Washington University Zebrafish Consortium Facility, and Fernanda Coelho for assistance with in situ hybridization. Crystal diffraction and SAXS data were collected at the APS, GM/CA 23-ID-D (supported by NCI ACB-12002 and the NIGMS AGM-12006 grants) and BioCAT 18-ID (supported by NIH grant 9 P41 GM103622-18), respectively. This research used resources of the APS, a U.S. Department of Energy (DOE) Office of Science User Facility operated for the DOE Office of Science by ANL under Contract No. DE-AC02-06CH11357. The Eiger 16 M detector was funded by an NIH–Office of Research Infrastructure Programs, High-End Instrumentation Grant (1S10OD012289-01A1). Use of the Pilatus 3 1 M detector was provided by grant 1S10OD018090-01 from NIGMS. This work was supported by NIH grants R01-GM120322 (D.A.), R01-NS079445 (K.R.M.), and T32GM007183 (K.L.). R.L.C. was supported by the National Science Foundation Graduate Research Fellowship (DGE-1745038). The content is solely the responsibility of the authors and does not necessarily reflect the official views of the National Institute of General Medical Sciences or the National Institutes of Health.

## Author contributions

K.L. cloned, expressed, purified proteins (with assistance from E.F.), carried out bioinformatic and biochemical characterizations, performed crystallography experiments (with assistance from J.L.) and structure determination, performed cAMP-based signaling assays, and collected and analyzed negative-stain EM data (with assistance from M.Z.). R.L.C. and K.R.M. designed and analyzed and R.L.C. performed zebrafish experiments. J.A.R., T.R.S., and K.L. designed and performed SAXS experiments. K.L. and D.A. designed all experiments, interpreted results, and wrote the manuscript.

## Competing interests

The authors declare no competing interests.
