## [Peer Review File · Nature Communications]

Reviewers' comments:

Reviewer #1 (Remarks to the Author):

Leon et al. describe the structural characterisation of the extracellular region of the zebrafish adhesion GPCR Gpr126. They have uncovered a new SEA domain, and a calcium-binding site at the N-terminal CUB domain that appears to have two functions: to maintain the closed structure of the inactive protein and to enable ligand binding and downstream signalling when the ECR has an open structure. They have shown that the presence of an alternatively spliced short exon is associated with the active open conformation.

These results will be of wide interest to the GPCR drug discovery field, developmental biologists—particularly researchers interested in myelination and inner ear formation—and clinicians, as human GPR126 mutations are associated with idiopathic scoliosis, arthrogryposis multiplex congenita and more widely with other diseases including cancer.

Overall, the manuscript is very interesting and has the potential to add significant new insight to the adhesion GPCR field. However, the arguments are not always clear, and there are several points that should be addressed before publication.

1. Throughout the text, the authors switch between human GPR126 and zebrafish Gpr126, but do not always clarify which species they are referring to, or stick to conventions for gene and protein nomenclature. It would help the reader if this was made clearer throughout the text. Examples include (but are not limited to):

a. Abstract, line 30: Gpr126/ADGRG6 mixes convention for the zebrafish and human proteins with the old and new names.

b. Line 32 should specify zebrafish as the species used.

c. Care needs to be taken with referencing work done in different species—for example the Geng et al. study and ear phenotype (line 111) refers to work on the zebrafish, while the Waller-Evans et al. study was done in the mouse (line 112).

Although one option would be to use zf and h, as in the figures (which are clear), this is not the accepted convention, and it would therefore be best to mention each species and use current conventions e.g. Gpr126 (zebrafish protein), GPR126 (human protein) in the main body of the text.

2. The authors use the negative stain EM data to suggest that the open conformation (vi) is not present in the -ss isoform, but in Figure 3E the 3rd sample from the left looks more open than class (iv). Is there a defined angle or threshold for the open or shut configuration?

3. Figure 4 demonstrates the effect of abolishing the calcium binding site in the -ss and +ss isoforms. Here it would aid understanding of the different isoform configurations if the supplementary data in Fig. S4E were included in the main Figure 4, with a schematic for each of the four different conformations (+ss,+Ca; -ss,+Ca; +ss,-Ca; -ss,-Ca) (combined 3F and S4E diagram).

4. In Figure S7 (and line 462), the alternative splicing of the CUB domain is introduced and this also needs a citation or data to support this scenario.

5. The authors assert that the CUB/PTX domain is required for heart development in the zebrafish. So far, this is not well supported by the literature and needs to be stated more cautiously. The authors discuss the heart defect in zebrafish gpr126 mutants with equality to the myelination and inner ear defects. However, the heart defect in zebrafish is based on a single morpholino and overexpression study (Patra et al., 2013) and therefore it is still possible that this is due to off-target/toxicity effects. One additional piece of evidence is that heart oedema was observed in maternal-zygotic stl47 mutants (Petersen et al., 2015), although this was only shown in a single supplementary image. This was work from the lab of Monk, whom we note is an author of the current study, but this observation is not cited in the context of the heart phenotype here.

It would therefore be of interest to know if the stl464 mutants described in the manuscript have a heart defect, which might be predicted if the CUB domain is no longer active. In fact, the fish in Figure 5D appears to have a slight cardiac oedema—is this a fully penetrant phenotype and have the authors characterised this further? Does this resemble the heart phenotype in adgrg6 morphants?

In relation to this point, the Geng et al. (2013), Monk et al. (2009) and Paavola et al. (2014) citations should all be removed from the list on line 354. Geng et al. (2013) showed expression, but not function, of gpr126 in the developing zebrafish heart, whereas the Monk et al. study did not have any mention of either cardiac expression or function. The Paavola et al. study noted normal heart development in zygotic and maternal-zygotic st86 mutants, but did not provide any supporting evidence for a role for zebrafish Gpr126 in heart development. In addition, the statement on line 469 ‘which has been shown...’ should be removed or toned down.

6. Have the authors tested if the adgrg6 transcript is present in the stl464 mutants to know if the protein is likely to be translated?

7. The overlay of the structures in Figure 6B make it difficult to see the conservation between the structures – it would be helpful if they could also be shown separately.

8. Presentation and discussion of cleavage by furin is ambiguous and contradictory in several places. The authors state on line 134 and in Fig. 6 that the SEA domain in the zebrafish protein is the site of furin-mediated cleavage, but elsewhere state that the zebrafish Gpr126 protein has ‘no furin cleavage site’ (line 154) and that ‘furin cleavage is not conserved in zebrafish Gpr126’ (line 370). However, the Discussion at lines 443 and 458-458 again mentions furin cleavage of Gpr126 (presumably zebrafish – this needs clarifying). In Fig. S6, a structure is shown which appears to be the zebrafish protein, marking a furin cleavage site, whereas the graphs in this figure relate to assays using the human protein. It would be helpful to have a paragraph in Discussion to be clear about furin cleavage/lack of in the zebrafish protein. It is important to be as clear as possible about this point, as Fig. 7E indicates that zebrafish Gpr126 is cleaved by furin to release the shorter ECR fragment and that this may be required for heart development (see also point 5 above).

9. Statistical tests should be provided for the graphed data shown in Figs 4, S4 and S6. A t-test is not suitable for comparison of more than two datasets. An ANOVA, with an appropriate post-test for multiple comparisons, should be used instead.

Minor points

1. The Discussion is repetitive e.g. lines 409/10 repeat the paragraph above.
2. Reference needed line 419

Reviewer #2 (Remarks to the Author):

The manuscript by Leon et al describes new mechanistic insights into the adhesion G protein coupled receptor GPR126 derived from a crystal structure of the zebrafish receptor ectodomain. Adhesion receptors remain one of the most poorly understood subfamilies of GPCRs, despite their critical demonstrated importance in a number of core developmental and physiological programs. The Arac-Ozkan lab has been leading the way for the past five years in elucidating how these receptors work. The structure presented in this manuscript describes the greatest structural insight into one of these families members to date.

Key strengths of the work include an integrative approach using mutagenesis studies, negative stain EM, evolutionary analysis, and knock-in zebrafish to understand which interactions observed in a high resolution crystal structure are critical for GPR126 function. A few very interesting results are proposed. First, splicing variation drives heterogeneity in the structure of the GPR126 ectodomain; this leads to increased basal signaling. Second, a conserved calcium site in the CUB domain is shielded in the basal state, but is critical for activation - mutation in both splice variants decreases signaling. Finally, the SEA domain of GPR126 likely bears a furin cleavage site, which may be important for unique activation mechanisms. Overall, the work presented here is of high quality and certainly deserves publication in Nature Communications. Some concerns that need to be addressed prior to publication:

1) The role of the splicing site, and potential glycosylations, are mentioned in the text, but it is unclear whether the protein crystallized is glycosylated, and if not, what the potential impacts for the observed heterogeneity in EM studies may be. One could imagine that this could be incredibly important, and the material from insect cells may not recapitulate function here adequately. Some caveats or discussion around this is required.

2) It's a bit surprising that analysis of the most novel domain domain in the structure (SEA) is restricted to modeling and a proposed cleavage site. Do mutants in cleavage alter signaling? I imagine the authors will follow up on some of these studies in zebrafish in future studies, but some validation of the proposed cleavage site would provide much needed confidence that this is not simply a structural fold.

Minor:

Fig 2B - Y535 is pointing a phenylalanine, F533 is pointing to a tyrosine

Fig 5B - nucleotide missing in red for the mutant

Pg 7, ln 169 - not clear why the analysis is suggested, but not presented.

Fig S1E, S2A - please indicate what kind of maps are presented in the legend

Reviewer #3 (Remarks to the Author):

The authors present a manuscript detailing the crystal structure of the extra-cellular region of a G-protein coupled receptor (Gpr126). The structure is demonstrated to be compact with unexpected domain contacts between regions well separated in the primary sequence. Further work is conducted using electron microscopy, small-angle X-ray scattering and other biophysical/biochemical methods to explain the potential impact of the closed conformation on function, and to contrast this with a more extended conformation observed in an alternatively spliced variant. The importance of a calcium binding domain is also highlighted.

The data supports the conclusions made and is in general a very well composed piece of work. Several minor things should be addressed prior to publication:

1. The experimental set-up and measured SAXS data should be summarised in a table according to the latest recommendations of the joint taskforce on scattering methods (Trehwella, Jill, et al. "2017 publication guidelines for structural modelling of small-angle scattering data from biomolecules in solution: an update." *Acta Crystallographica Section D: Structural Biology* 73.9 (2017): 710-728.)
2. SAXS data and any corresponding models (including fits) should be deposited in the publicly available and curated database www.sasbdb.org, as also recommended in the aforementioned recommendations.
3. It is not clear what model was used to describe the "extended conformation" and extract a theoretical R_g . Please include this model and make it clear.
4. As the authors indicate that conformational dynamics are likely essential for the function of the ECR, and that the authors have obtained high quality SAXS data, the structure determined and SAXS data should be used to examine this. It is suggested that an ensemble analysis is performed (using eg. EOM from the ATSAS suite - Tria, Giancarlo, et al. "Advanced ensemble modelling of flexible macromolecules using X-ray solution scattering." *IUCrJ* 2.2 (2015): 207-217). The distributions of distances determined would be very illuminating and further the impact of this paper.
5. On page 8, the paragraph from lines 186-192 is confusing. Please restructure for clarity.
6. Please reference the corresponding beamline papers in the methods for crystallography and scattering where available.
7. As real-space distance distributions (Fourier transforms of the SAXS data) are not shown in the paper it is unnecessary to include the corresponding fits to the data. Please either include these distributions or remove the fits from the SAXS figures. The fits make it hard to appreciate the differences in the measured SAXS data for the +ss and -ss samples.

8. According to the recommendations for scattering data publication, please include a panel for each data set showing residuals and/or errors. Even though the SAXS data only appears in the supplementary section it is still important that the reader can appreciate if differences are significant.

Reviewer #4 (Remarks to the Author):

The authors present a set of structural, pharmacological and in vivo assays using zebrafish and uncover novel aspects of the adhesion GPCR Gpr126. The authors are experts in the analysis of this receptor class and the manuscript contains the description of several novel, insightful features of this interesting receptor group. Foremost, the identification of hitherto unknown SEA domain in the ECR of Gpr126 is interesting, and the structure-based evidence on different ECR conformations that may result from alternative splicing and calcium binding are of high and general interest. Unfortunately, the manuscript is partly confusing and imprecise when it comes to the integration of structural and pharmacological datasets, and completely lacks evidence for speculations raised on the effects of the identified structural properties of the ECR and ligand engagement. The authors need to provide further evidence in order to substantiate their claims how structure impacts on the function of the receptor in the context of ligand and calcium binding. In the current form I cannot recommend publication of the manuscript, but I have laid out pivotal experiments that I would like to see in order to underpin and bridge the solid structural and physiological datasets with sound pharmacological evidence.

MAJOR POINTS

p8, 177

The hLPHN3 GAIN structure was not published in Arac et al., 2012 so the reference is misplaced and should be moved before hLphn3 in the sentence.

Further, I was astounded to find the hLPHN3 GAIN domain displayed in this manuscript uncommented. There is no explanation as to whether the rendering displayed in Fig. S1F is based on a homology model (if so, a short section in the methods section on how was this done is required) or a novel X-ray crystallographic structure (if so, a PDB ID, the entire experimental description of its production and solution and short discussion in the text is required).

Figures 4A-C, S4B-D, S6B-D

Error bars are missing in the diagrams in 4A, S4B and S6B. Please provide those along with a statistical measure testing whether the displayed groups differ in surface expression levels from each other.

p12

The authors provide compelling structural evidence for the importance of the calcium binding site in Gpr126 through mutagenesis approaches, which remove the calcium binding site. These datasets are very interesting and central to establish structure-function correlates and ultimately causality between the observed features of pr126. However, I feel that the authors need to focus more diligently on the interconnection and presentation of the many facets they have uncovered in this manuscript, and correct or further substantiate some of the claims they make here:

- This set of experiments shows that the calcium binding site bound to calcium stabilizes the closed conformation that is commonly observed in the -ss form, because mutations to the site result in an opening of the conformation adopting a +ss-like shape. In a second line of experiments the authors first establish that the -ss and +ss have different signaling effects that can be summarized as follows: -ss/closed conformation/reduced cAMP signal as opposed to +ss/open conformation/elevated cAMP signal.

- Here, the logic as to how calcium binding factors into this framework, is ill-explained or ill-founded. The authors show that removal of the calcium binding site in the -ss form results in an open conformation as expected if the site stabilized the closed conformation of the ECR. However, no change in cAMP signaling was observed. In the contrary, the already opened +ss conformation suffered through the calcium site mutations in signaling strength. If the calcium binding site stabilizes the low-signaling -ss closed conformation of the receptor, then mutations of the site should have a stimulating effect on the -ss form and no effect on the high-signaling open +ss form in signaling assays. But the authors document the inverse situation.

Therefore, the sentence 'These results suggest that the calcium-binding site is important for increased signaling in the (+ss) isoform and is likely a functional site for Gpr126.' (p13, 311-313) is true but does not reflect or is even opposing the findings on the conformational role of the calcium

binding site and the impact of conformational changes on receptor signaling such as proposed by the authors above.

- Further, there is no mention and no experimental evidence providing an explanation as to why the calcium binding site may have a third function, additional to the closed conformation stabilization and cAMP signal suppression: as a binding interface to ligands. Only on p17 (414-417) the authors introduce this concept in the discussion: 'Thus, we speculate that Gpr126 uses this site to not only bind to itself in a closed conformation (-ss isoform), effectively hiding the calcium-binding site from ligands, but also to bind to ligands when the ECR is in a possible open conformation (+ss isoform) in order to activate signaling.'

As the manuscript does not contain any experimental effort to test for differing ligand interactions between the -ss and +ss isoforms, nor the effect of abolishing the calcium binding site on ligand engagement and signaling, this statement is a clear overinterpretation of the data that are currently provided. Gpr126 has three identified ligands and none was tested in this manuscript. This could have been performed by the authors given their superb protein biochemical and genetic expertise, and backed by the fact that some of the co-authors were involved in the identification of two of the Gpr126 ligands (laminin-211, PrP-C). Without such testing, the invention of a ligand-binding role for the calcium binding site is premature, confusing and thus misleading. Thus, the authors should provide evidence that abolishment of the calcium binding mode of the Gpr126 ECR is reflected changes in ligand binding, and signaling.

- If closed and open ECR conformations represent two signaling states of Gpr126, and if this transition can be influenced by calcium as a factor that stabilizes the closed conformation of the receptor, this process should be physiologically prone to changes in calcium level and likely dynamic. These changes may occur either individually or simultaneously in the ER/Golgi/secretory pathway and/or in the extracellular space. Calcium levels can be easily manipulated in *in vitro* signaling assays as conducted by the authors by changing extracellular calcium levels through buffers or by changing the composition of the medium. Intracellular and intra-ER calcium levels can be manipulated through the use of SERCA inhibitors such as thapsigargin. This will allow to test whether calcium changes impinge on signaling outcome of the -ss and +ss forms, and should be abolished in the calcium binding site mutants of either receptor isoform substantiating the speculation that conformational transitions between closed and open ECR conformation are attributable for this effect. Note: I am not asking for further structural studies under different calcium conditions as I am aware of the substantial efforts required to conduct these assays. But at least one independent line of evidence should underpin the speculation that the calcium binding site identified in Gpr126 regulates signaling behavior of the receptor through its role in the ECR structure.

- The authors should mention and discuss why the tethered agonist peptide used in their Gpr126 stimulation assay differs significantly in size from the previously published one in Liebscher et al., 2014. The authors here used a p7 peptide, while in the original paper a p16 was identified as the

most potent one from a library of gradually elongated tethered agonist portions and used to stimulate activity of Gpr126. This seems important as the p7 peptide reported in Liebscher et al. did not affect cAMP accumulation over control conditions. Have the authors tested a p16 peptide as well, what was the outcome compared to the p7 variant?

p16, p390-392

The statement with regards to ligand binding ('the importance of furin cleavage is likely not primarily important for proper expression and trafficking but rather for a ligand-mediated function') has no experimental foundation. I would request to remove the statement regarding a potential ligand interaction through the SEA domain, or ask the authors to demonstrate evidence for ligand binding at the SEA domain in relation to furin processing (remove furin activity, remove furin cleavage site).

Fig. S6

The authors mention that they engineered a furin-cleavage resistant Gpr126 mutant but show no evidence for this effect on furin cleavage. The authors are asked to demonstrate cleavage-deficiency effected by the mutation, e.g. by a Western blot.

I assume that R468A represents this mutation? If so, this is neither mentioned in the main text nor the figure caption, but only in the diagrams in S6B-D. Insert a suitable explanation in main text and caption.

MINOR POINTS

p3, 67-68

There are only 32 human aGPCR genes currently known, EMR4 is considered a pseudogene in the human genome. Please correct.

p5, 120-121

'... 150 aa region between PTX and HormR, could not be identified.'

I'd suggest to rephrase since apparently the fact that there is a linker region between the said domains, to which a structure could not be readily assigned, was already known.

'... 150 aa region between PTX and HormR, could not be assigned to a known structural fold.'

Figure S4E

A grouping of observed conformations such as presented in Fig. 3D is missing for this dataset. Please provide.

Figure caption S4C

'Basal signaling for sam constructs ...'. Correct

p15, 370

Insert bold words in statement: 'Although **THE** furing cleavage **SITE** is not conserved.....'

p16, 388-389

There is a problem with grammar of this sentence:

'HEK293 cells transfected with a mutant form of human GPR126 that is not able to be cleaved by furin were detected on the cell surface...'

Change to:

'Human GPR126 that is not able to be cleaved by furin was transfected in HEK293 cells and detected on the cell surface...'

Fig. 7E

Is the ECR shed after furin cleavage? There is not evidence for this in this manuscript, and this would also be in contrast to data obtained from the Notch receptor field showing that furin cleavage results in a non-covalent heterodimer (similar to the SEA and GAIN domain autoproteolysis), which does not readily dissociate (see Logeat et al., PNAS, 1998; Blaumueller et al., Cell 1997, Rand et al., Mol Cell Biol, 2000).

Hence, the cartoon introduces a situation - NTF shedding - that is has not been formally experimentally associated with neither furin cleavage, SEA domain autoproteolysis nor GAIN domain self-cleavage. The latter point has been only partially addressed by artificially or genetically produced Gpr126 NTF, which can rescue Gpr126 hypomorphic phenotypes in vivo. However, to my knowledge it was not tested whether these phenotypes in heart development can be complemented by using a GAIN domain autoproteolysis-disrupting mutation in Gpr126. In sum, the scenario of NTF shedding is very speculative and further elaboration on this without concrete evidence is misleading readers.

Response to Reviewer Comments

Structural basis for adhesion G protein-coupled receptor Gpr126 function
NCOMMS-19-13915-T

Reviewer #1 (Remarks to the Author):

Leon et al. describe the structural characterisation of the extracellular region of the zebrafish adhesion GPCR Gpr126. They have uncovered a new SEA domain, and a calcium-binding site at the N-terminal CUB domain that appears to have two functions: to maintain the closed structure of the inactive protein and to enable ligand binding and downstream signalling when the ECR has an open structure. They have shown that the presence of an alternatively spliced short exon is associated with the active open conformation.

These results will be of wide interest to the GPCR drug discovery field, developmental biologists—particularly researchers interested in myelination and inner ear formation—and clinicians, as human GPR126 mutations are associated with idiopathic scoliosis, arthrogryposis multiplex congenita and more widely with other diseases including cancer.

Overall, the manuscript is very interesting and has the potential to add significant new insight to the adhesion GPCR field. However, the arguments are not always clear, and there are several points that should be addressed before publication.

→ We thank the reviewer for his/her enthusiasm and constructive comments. We addressed all comments as below:

1. Throughout the text, the authors switch between human GPR126 and zebrafish Gpr126, but do not always clarify which species they are referring to, or stick to conventions for gene and protein nomenclature. It would help the reader if this was made clearer throughout the text. Examples include (but are not limited to):

a. Abstract, line 30: Gpr126/ADGRG6 mixes convention for the zebrafish and human proteins with the old and new names.

b. Line 32 should specify zebrafish as the species used.

c. Care needs to be taken with referencing work done in different species—for example the Geng et al. study and ear phenotype (line 111) refers to work on the zebrafish, while the Waller-Evans et al. study was done in the mouse (line 112).

Although one option would be to use zf and h, as in the figures (which are clear), this is not the accepted convention, and it would therefore be best to mention each species and use current conventions e.g. Gpr126 (zebrafish protein), GPR126 (human protein) in the main body of the text.

→ We thank the reviewer for pointing this out. We made all the suggested changes and have taken care to make the distinctions between human and zebrafish clearer in the main text.

2. The authors use the negative stain EM data to suggest that the open conformation (vi) is not present in the -ss isoform, but in Figure 3E the 3rd sample from the left looks more open than class (iv). Is there a defined angle or threshold for the open or shut configuration?

→ To distinguish between open and closed conformations for particles which have the “V-shape” conformation, we are using a threshold of 50°. This threshold is slightly larger than the angle observed in the Gpr126 -ss ECR crystal structure model and accounts for the dynamic properties of proteins rather than the static depictions observed through crystallography. We

have modified the figure legend to clarify this threshold. The 3rd sample from the left in Figure 3E appears to be ~50°, around the threshold.

3. Figure 4 demonstrates the effect of abolishing the calcium binding site in the -ss and +ss isoforms. Here it would aid understanding of the different isoform configurations if the supplementary data in Fig. S4E were included in the main Figure 4, with a schematic for each of the four different conformations (+ss,+Ca; -ss,+Ca; +ss,-Ca; -ss,-Ca) (combined 3F and S4E diagram).

→ We have now included cartoon representations of the constructs used in the signaling assay in Supplementary Figure 4G (note: because this signaling assay was moved to Supplementary Figure 4H, it is no longer in Figure 4).

4. In Figure S7 (and line 462), the alternative splicing of the CUB domain is introduced and this also needs a citation or data to support this scenario.

→ We have now included the citation in this sentence.

5. The authors assert that the CUB/PTX domain is required for heart development in the zebrafish. So far, this is not well supported by the literature and needs to be stated more cautiously. The authors discuss the heart defect in zebrafish *gpr126* mutants with equality to the myelination and inner ear defects. However, the heart defect in zebrafish is based on a single morpholino and overexpression study (Patra et al., 2013) and therefore it is still possible that this is due to off-target/toxicity effects. One additional piece of evidence is that heart oedema was observed in maternal-zygotic *stl47* mutants (Petersen et al., 2015), although this was only shown in a single supplementary image. This was work from the lab of Monk, whom we note is an author of the current study, but this observation is not cited in the context of the heart phenotype here.

It would therefore be of interest to know if the *stl464* mutants described in the manuscript have a heart defect, which might be predicted if the CUB domain is no longer active. In fact, the fish in Figure 5D appears to have a slight cardiac oedema—is this a fully penetrant phenotype and have the authors characterised this further? Does this resemble the heart phenotype in *adgrg6* morphants?

→ We thank the reviewer for raising this point. We agree that the importance of CUB/PTX in heart development in zebrafish is not as well supported by the literature and have made changes in the main text to state this more cautiously. We have also included the Petersen et al. (2015) citation, as the reviewer pointed out.

We have examined the heart phenotypes of the *stl464* mutant zebrafish and observed no heart defects, which suggests that the calcium-binding site of *Gpr126* is not important for a heart development function in zebrafish. We have included these images in Supplementary Figure 5E-T and modified the main text to include these results. However, we note that we cannot exclude the possibility that the CUB domain is important for a heart development function, as there may be additional unidentified functional sites distinct from the calcium-binding site that are still intact and important for this function.

In relation to this point, the Geng et al. (2013), Monk et al. (2009) and Paavola et al. (2014) citations should all be removed from the list on line 354. Geng et al. (2013) showed expression, but not function, of *gpr126* in the developing zebrafish heart, whereas the Monk et al. study did

not have any mention of either cardiac expression or function. The Paavola et al. study noted normal heart development in zygotic and maternal-zygotic st86 mutants, but did not provide any supporting evidence for a role for zebrafish Gpr126 in heart development. In addition, the statement on line 469 'which has been shown...' should be removed or toned down.

→ We apologize for the confusing sentence. The Geng et al. (2013), Monk et al. (2009), and Paavola et al. (2014) citations were referring to the myelination and ear development functions stated earlier in the same sentence and should have been placed more appropriately. However, we have removed this paragraph altogether due to the new data that shows that the stl464 mutants do not have heart defects.

We have also removed the statement in line 469.

6. Have the authors tested if the adgrg6 transcript is present in the stl464 mutants to know if the protein is likely to be translated?

→ We thank the reviewer for this important question. We have carried out this experiment and found that the adgrg6 transcript is present in the stl464 mutants, similar to the wild-type fish. This suggests that the transcript is likely translated and that the mutant phenotype is not due to a disruption in gene expression. We have included these data in Supplementary Figure 5C-D.

7. The overlay of the structures in Figure 6B make it difficult to see the conservation between the structures – it would be helpful if they could also be shown separately.

→ We have included the separate structures in this figure.

8. Presentation and discussion of cleavage by furin is ambiguous and contradictory in several places. The authors state on line 134 and in Fig. 6 that the SEA domain in the zebrafish protein is the site of furin-mediated cleavage, but elsewhere state that the zebrafish Gpr126 protein has 'no furin cleavage site' (line 154) and that 'furin cleavage is not conserved in zebrafish Gpr126' (line 370). However, the Discussion at lines 443 and 458-458 again mentions furin cleavage of Gpr126 (presumably zebrafish – this needs clarifying). In Fig. S6, a structure is shown which appears to be the zebrafish protein, marking a furin cleavage site, whereas the graphs in this figure relate to assays using the human protein. It would be helpful to have a paragraph in Discussion to be clear about furin cleavage/lack of in the zebrafish protein. It is important to be as clear as possible about this point, as Fig. 7E indicates that zebrafish Gpr126 is cleaved by furin to release the shorter ECR fragment and that this may be required for heart development (see also point 5 above).

→ We thank the reviewer for pointing out this confusing discussion of the furin cleavage. We have modified the main text to state more clearly that any mentions of furin cleavage refer to the human protein and that this cleavage site is not conserved in zebrafish. In Supplementary Figure 6A, we are showing the zebrafish protein but we expect that the human protein would look very similar due to moderate sequence identity (47%), with some small differences such as a longer loop in the human SEA domain that accommodates the furin cleavage site. We have modified this figure to make it clear that the structure is zebrafish, but we are highlighting the corresponding furin cleavage site found in human.

9. Statistical tests should be provided for the graphed data shown in Figs 4, S4 and S6. A t-test is not suitable for comparison of more than two datasets. An ANOVA, with an appropriate post-test for multiple comparisons, should be used instead.

→ We thank the reviewer for this correction and have performed one-way ANOVA tests with Tukey's multiple comparisons test for the signaling assay results in Figures 4 and S4. The signaling assay from Figure S6 was removed due to unreliable expression levels (see Reviewer 2, Comment 2).

Minor points

1. The Discussion is repetitive e.g. lines 409/10 repeat the paragraph above.
2. Reference needed line 419

→ We have removed the repetitive sentence and included the appropriate references.

Reviewer #2 (Remarks to the Author):

The manuscript by Leon et al describes new mechanistic insights into the adhesion G protein coupled receptor GPR126 derived from a crystal structure of the zebrafish receptor ectodomain. Adhesion receptors remain one of the most poorly understood subfamilies of GPCRs, despite their critical demonstrated importance in a number of core developmental and physiological programs. The Arac-Ozkan lab has been leading the way for the past five years in elucidating how these receptors work. The structure presented in this manuscript describes the greatest structural insight into one of these families members to date.

Key strengths of the work include an integrative approach using mutagenesis studies, negative stain EM, evolutionary analysis, and knock-in zebrafish to understand which interactions observed in a high resolution crystal structure are critical for GPR126 function. A few very interesting results are proposed. First, splicing variation drives heterogeneity in the structure of the GPR126 ectodomain; this leads to increased basal signaling. Second, a conserved calcium site in the CUB domain is shielded in the basal state, but is critical for activation - mutation in both splice variants decreases signaling. Finally, the SEA domain of GPR126 likely bears a furin cleavage site, which may be important for unique activation mechanisms. Overall, the work presented here is of high quality and certainly deserves publication in Nature Communications. Some concerns that need to be addressed prior to publication:

→ We thank the reviewer for his/her kind and enthusiastic comments about our work and appreciate the suggestions.

1) The role of the splicing site, and potential glycosylations, are mentioned in the text, but it is unclear whether the protein crystallized is glycosylated, and if not, what the potential impacts for the observed heterogeneity in EM studies may be. One could imagine that this could be incredibly important, and the material from insect cells may not recapitulate function here adequately. Some caveats or discussion around this is required.

→ We agree with the reviewer that glycosylation is an important factor to consider and discuss in this work. The crystallized (-ss isoform) protein produced in insect cells is indeed glycosylated, as we observe density at 10 of the 15 predicted N-linked glycosylation sites. All other proteins in this study should also be glycosylated, including the +ss splice isoform. This isoform has one more N-linked glycosylation and likely other O-linked glycosylation sites in its extended linker region, which may contribute to the differences in ECR conformation between the two isoforms by introducing additional bulk, which could prevent a stable closed ECR conformation.

We also agree that proteins produced in insect cells have simpler glycosylation patterns and do not fully represent the native proteins expressed in zebrafish and/or human. However, we believe that, despite the production in insect cells, the structural differences between the two splice isoforms are supported by several observations:

- 1) Differences in ECR conformation between zebrafish Gpr126 -ss and +ss isoforms by negative stain EM (proteins produced in insect cells)
- 2) Differences in R_g and D_{max} between zebrafish Gpr126 -ss and +ss isoforms as well as between human GPR126 -ss and +ss isoforms by SAXS (proteins produced in insect cells)
- 3) In addition, differences in cAMP signaling between zebrafish Gpr126 -ss and +ss isoforms as well as between human GPR126 -ss and +ss isoforms (proteins produced in HEK293 cells) suggest a functional distinction between the splice isoforms

Despite the simpler glycosylation pathways in insect cells, we observe structural differences between the -ss and +ss isoforms through both negative stain EM and SAXS. We predict that the same proteins produced in mammalian cells would result in more dramatic structural changes between splice isoforms due to the longer and more complex glycosylation chains.

We have modified the main text to clarify the glycosylation status of the crystallized protein and the potential contribution of glycosylation to the observed differences in ECR conformation.

[Redacted]

Minor:

Fig 2B - Y535 is pointing a phenylalanine, F533 is pointing to a tyrosine

Fig 5B - nucleotide missing in red for the mutant

Pg 7, Ln 169 - not clear why the analysis is suggested, but not presented.

Fig S1E, S2A - please indicate what kind of maps are presented in the legend

→ We thank the reviewer for his/her careful reading of the manuscript and figures and have made corrections for all of the above points.

Reviewer #3 (Remarks to the Author):

The authors present a manuscript detailing the crystal structure of the extra-cellular region of a G-protein coupled receptor (Gpr126). The structure is demonstrated to be compact with unexpected domain contacts between regions well separated in the primary sequence. Further work is conducted using electron microscopy, small-angle X-ray scattering and other biophysical/biochemical methods to explain the potential impact of the closed conformation on function, and to contrast this with a more extended conformation observed in an alternatively spliced variant. The importance of a calcium binding domain is also highlighted.

The data supports the conclusions made and is in general a very well composed piece of work. Several minor things should be addressed prior to publication:

→ We thank the reviewer for his/her enthusiasm and helpful suggestions.

1. The experimental set-up and measured SAXS data should be summarised in a table according to the latest recommendations of the joint taskforce on scattering methods (Trehwella, Jill, et al. "2017 publication guidelines for structural modelling of small-angle scattering data from biomolecules in solution: an update." *Acta Crystallographica Section D: Structural Biology* 73.9 (2017): 710-728.)

→ We have added the requested table in Supplementary Table 3.

2. SAXS data and any corresponding models (including fits) should be deposited in the publicly available and curated database www.sasbdb.org, as also recommended in the aforementioned recommendations.

→ We have deposited the SAXS data, and the depositions can be accessed at the following links:

<https://www.sasbdb.org/data/SASDFT9/11qel20crz/>

<https://www.sasbdb.org/data/SASDFU9/caeea7zgbw/>

<https://www.sasbdb.org/data/SASDFV9/s5rji8qof8/>

<https://www.sasbdb.org/data/SASDFW9/gqx5esjbhr/>

<https://www.sasbdb.org/data/SASDFX9/83mzf5qa5t/>

3. It is not clear what model was used to describe the "extended conformation" and extract a theoretical R_g . Please include this model and make it clear.

→ As requested, we have added an image of the extended model to Supplementary Figure 1G. Although the carbohydrates are not shown, the scattering profile has been calculated with them, as described in the methods using ATSAS tools.

4. As the authors indicate that conformational dynamics are likely essential for the function of the ECR, and that the authors have obtained high quality SAXS data, the structure determined and SAXS data should be used to examine this. It is suggested that an ensemble analysis is performed (using eg. EOM from the ATSAS suite - Tria, Giancarlo, et al. "Advanced ensemble modelling of flexible macromolecules using X-ray solution scattering." IUCrJ 2.2 (2015): 207-217). The distributions of distances determined would be very illuminating and further the impact of this paper.

→ We thank the reviewer for this suggestion. We ran EOM as requested (Figure X2). The results, however, are compromised as EOM does not allow for the inclusion of the ~25kD carbohydrates (which do significantly contribute to the scattering: the X-ray contrast of sugars, which are ~20% of the total mass of our system, is about 50% higher than for proteins). Hence, we prefer not to include this analysis in the paper. We acknowledge more analysis is beneficial and include the $P(r)$ calculation in Supplementary Figure 3C-D.

Figure X2. Ensemble Optimization Method (EOM) analysis of zebrafish Gpr126 splice isoforms.

- (A) Distributions of R_g for the pool ensembles generated by EOM analysis for zebrafish -ss (black) and +ss (red) ECRs.
(B) Distributions of D_{max} for the pool ensembles generated by EOM analysis for zebrafish -ss (black) and +ss (red) ECRs.

5. On page 8, the paragraph from lines 186-192 is confusing. Please restructure for clarity.

→ We have edited this section for clarity as suggested.

6. Please reference the corresponding beamline papers in the methods for crystallography and scattering where available.

→ Both GM/CA and BioCAT do not list any publications to be cited, but we have included all of the acknowledgement statements provided on their websites. We have also included the beamline information in both methods and acknowledgements.

7. As real-space distance distributions (Fourier transforms of the SAXS data) are not shown in the paper it is unnecessary to include the corresponding fits to the data. Please either include these distributions or remove the fits from the SAXS figures. The fits make it hard to appreciate the differences in the measured SAXS data for the +ss and -ss samples.

→ We thank the reviewer for this comment. We now include a plot of the $P(r)$ in Supplementary Figure 3C-D. We denote it as $P^{app}(r)$ to emphasize that it is for a mixed protein/carbohydrate system and hence, the common interpretation that it is the distance distribution of vector lengths is invalid (the vectors should be weighted by the electron contrast of the molecule).

8. According to the recommendations for scattering data publication, please include a panel for each data set showing residuals and/or errors. Even though the SAXS data only appears in the supplementary section it is still important that the reader can appreciate if differences are significant.

→ This information is now included in Supplementary Table 3.

Reviewer #4 (Remarks to the Author):

The authors present a set of structural, pharmacological and in vivo assays using zebrafish and uncover novel aspects of the adhesion GPCR Gpr126. The authors are experts in the analysis of this receptor class and the manuscript contains the description of several novel, insightful features of this interesting receptor group. Foremost, the identification of hitherto unknown SEA domain in the ECR of Gpr126 is interesting, and the structure-based evidence on different ECR conformations that may result from alternative splicing and calcium binding are of high and general interest. Unfortunately, the manuscript is partly confusing and imprecise when it comes to the integration of structural and pharmacological datasets, and completely lacks evidence for speculations raised on the effects of the identified structural properties of the ECR and ligand engagement. The authors need to provide further evidence in order to substantiate their claims how structure impacts on the function of the receptor in the context of ligand and calcium binding. In the current form I cannot recommend publication of the manuscript, but I have laid out pivotal experiments that I would like to see in order to underpin and bridge the solid structural and physiological datasets with sound pharmacological evidence.

→ We thank the reviewer for his/her constructive comments and for suggesting a path to improve the manuscript. We agree with the reviewer and understand his/her concerns. We believe we addressed all concerns and changed the manuscript accordingly. We present our answers/new results and other discussion below. We apologize for the lengthy discussion below. In summary:

-We performed ligand binding experiments and signaling assays with PrP^C and type IV collagen and were unable to detect reliable binding or effects on receptor signaling.

-We performed the suggested experiments using EGTA and thapsigargin and obtained unreliable results.

-Thus, we shuffled our calcium-binding mutant data to a supplementary figure rather than a main figure and changed the model in Figure 7 and have removed our claims about the calcium-binding site being a ligand-binding site.

MAJOR POINTS

p8, 177

The hLPHN3 GAIN structure was not published in Arac et al., 2012 so the reference is misplaced and should be moved before hLphn3 in the sentence.

Further, I was astounded to find the hLPHN3 GAIN domain displayed in this manuscript uncommented. There is no explanation as to whether the rendering displayed in Fig. S1F is based on a homology model (if so, a short section in the methods section on how was this done is required) or a novel X-ray crystallographic structure (if so, a PDB ID, the entire experimental description of its production and solution and short discussion in the text is required).

→ The hLPHN3 GAIN structure is a novel X-ray crystallographic structure from our lab that is currently unpublished. We have removed it from the figure to avoid confusion.

Figures 4A-C, S4B-D, S6B-D

Error bars are missing in the diagrams in 4A, S4B and S6B. Please provide those along with a statistical measure testing whether the displayed groups differ in surface expression levels from each other.

→ We acknowledge the lack of error bars in these figures and hope this will offer clarification: The data in the referenced figures show one signaling assay that is representative of at least three separate experiments. Each experiment includes a signaling assay and a cell-surface expression assay performed on the same transfected population of cells to report cell-surface expression level that directly matches to the cells that were used in the particular signaling assay. Our method for measuring cell-surface expression is to stain transfected cells (in parallel with cAMP measurement) with antibodies (1°: mouse M2 anti-FLAG, 2°: donkey anti-mouse Alexa Fluor 488) and measure the median fluorescence intensity of the cells using flow cytometry. Because we measure fluorescence intensity values for numerous cells (~10,000), we take the median fluorescence intensity as the “cell-surface expression” value that is shown in the figures. Therefore, for each experiment we are using a single, median expression value for each population of transfected cells and do not have error bars for these values. However, please see Figure X3 for expression data averaged among triplicate experiments, showing consistent expression levels for all constructs. We have also included raw flow cytometry data in Supplementary Figure 4A-B.

Figure X3. Cell-surface expression analysis of Gpr126/GPR126 constructs.

(A) Cell-surface expression for HEK293 cells transfected with empty vector (EV) and FLAG-tagged zebrafish Gpr126 constructs, averaged over three independent experiments.

(B) Cell-surface expression for HEK293 cells transfected with empty vector (EV) and FLAG-tagged human GPR126 constructs, averaged over three independent experiments. Expression was measured by detecting fluorescent signals on stained cells (1° antibody: mouse M2 anti-FLAG, 2° antibody: donkey anti-mouse Alexa Fluor 488) by flow cytometry.

p12

The authors provide compelling structural evidence for the importance of the calcium binding site in Gpr126 through mutagenesis approaches, which remove the calcium binding site. These datasets are very interesting and central to establish structure-function correlates and ultimately causality between the observed features of pr126. However, I feel that the authors need to focus more diligently on the interconnection and presentation of the many facets they have uncovered in this manuscript, and correct or further substantiate some of the claims they make here:

→ We thank the reviewer for the positive comments on the calcium-binding site work and constructive comments that will help strengthen the manuscript.

- This set of experiments shows that the calcium binding site bound to calcium stabilizes the closed conformation that is commonly observed in the -ss form, because mutations to the site result in an opening of the conformation adopting a +ss-like shape. In a second line of experiments the authors first establish that the -ss and +ss have different signaling effects that can be summarized as follows: -ss/closed conformation/reduced cAMP signal as opposed to +ss/open conformation/elevated cAMP signal.

→ We agree with these interpretations of our work. We have now rearranged Figure 4 and Supplementary Figure 4. We now show in the main figure that alternative splicing has an effect on signaling. We moved the calcium-binding mutation data to the supplement.

- Here, the logic as to how calcium binding factors into this framework, is ill-explained or ill-founded. The authors show that removal of the calcium binding site in the -ss form results in an

open conformation as expected if the site stabilized the closed conformation of the ECR. However, no change in cAMP signaling was observed.

→ That is right.

In the contrary, the already opened +ss conformation suffered through the calcium site mutations in signaling strength.

→ That is right, too.

If the calcium binding site stabilizes the low-signaling -ss closed conformation of the receptor, then mutations of the site should have a stimulating effect on the -ss form and no effect on the high-signaling open +ss form in signaling assays.

→ This is the most reasonable and simplest model. However, the data does not fit to this model.

But the authors document the inverse situation.

→ That is right. We originally thought and still think that this data can be explained by a more complex (and yet ill-explained) model which is: In addition to the above simple model, activation also needs binding of a ligand to the calcium-binding site and that the ligand is always present in our experimental system (HEK cell cultures). In other words, the protein should be “open” and also “be able to bind to its ligand” simultaneously so that it can be active. Only the wild type +ss isoform meets these requirements, and thus only that protein is “more active” and the others are not.

Therefore, the sentence 'These results suggest that the calcium-binding site is important for increased signaling in the (+ss) isoform and is likely a functional site for Gpr126.' (p13, 311-313) is true but does not reflect or is even opposing the findings on the conformational role of the calcium binding site and the impact of conformational changes on receptor signaling such as proposed by the authors above.

→ We are aware that the simple model contradicts the data; and that the more complex and ill-explained model agrees with the data. For this reason, we prefer to move the mutant data to Supplementary Figure 4H and remove claims about it and do not mention the complex model. We believe this will address the reviewer's concern.

If interested: here is a more detailed explanation of what we think:

We agree that these sets of signaling data are confusing and seem to be contradictory to our argument that the calcium-binding site is important for cAMP signaling. We hope that the following explanation will provide better context for this argument. With regard to the observation that cells transfected with the +ss isoform have higher levels of cAMP compared to cells transfected with the -ss isoform, we believe this could be due to:

- 1) The change in ECR conformation from closed to more open results in differences in signaling, possibly by changing ECR-7TM regulatory interactions. This change is inherent to the ECR conformation itself and does not require additional factors.
- 2) The change in ECR conformation from closed to more open exposes the calcium-binding site which might act as a ligand-binding site and activate cAMP signaling.

If the first model is correct, we would expect to see an increase in signaling when the -ss isoform is mutated in the calcium-binding site, because the calcium-binding site contributes to a stable closed conformation. We would not expect a large signaling change when the +ss isoform is mutated in the calcium-binding site, because the calcium-binding site has less of a contribution to maintaining a closed conformation for this isoform, which is more open.

If the second model is correct, we would not expect to see a significant change in signaling when the -ss isoform is mutated in the calcium-binding site, since in both cases: a) wild-type -ss isoform and b) D134A/F135A calcium-binding site mutant of -ss isoform, the calcium-binding site is a) obscured or b) mutated. Hence, we believe the calcium-binding site would be “inactive” in both scenarios and would not signal to a higher level. We would, however, expect a decrease in signaling upon mutation of the calcium-binding site in the +ss isoform. The wildtype +ss, the ECR conformation is open and the calcium-binding site would be exposed to ligands in the extracellular matrix, which would lead to an increased signal (compared to the -ss isoform). Mutation of the calcium binding site in the +ss isoform would not affect the ECR conformation but it would disrupt the calcium-binding site such that ligands would not be able to bind there and activate signaling.

The signaling assay results suggest that the change in ECR conformation alone is not the reason for the higher signaling observed in the +ss isoform. We think that the role of the calcium-binding site is to act as a binding interface for at least two regulatory modes. The first, when the ECR conformation is closed, the calcium-binding site is used to bind to a distant site on Gpr126. This obscures the calcium-binding site from ligand binding events and keeps basal signaling at a lower level. The second, when the ECR conformation is open, the calcium-binding site is exposed to endogenous ligands and signaling can be activated to a higher level.

This is our explanation of the data. However, we did not include these claims in the manuscript.

- Further, there is no mention and no experimental evidence providing an explanation as to why the calcium binding site may have a third function, additional to the closed conformation stabilization and cAMP signal suppression: as a binding interface to ligands. Only on p17 (414-417) the authors introduce this concept in the discussion: 'Thus, we speculate that Gpr126 uses this site to not only bind to itself in a closed conformation (-ss isoform), effectively hiding the calcium-binding site from ligands, but also to bind to ligands when the ECR is in a possible open conformation (+ss isoform) in order to activate signaling.'

As the manuscript does not contain any experimental effort to test for differing ligand interactions between the -ss and +ss isoforms, nor the effect of abolishing the calcium binding site on ligand engagement and signaling, this statement is a clear overinterpretation of the data that are currently provided. Gpr126 has three identified ligands and none was tested in this manuscript. This could have been performed by the authors given their superb protein biochemical and genetic expertise, and backed by the fact that some of the co-authors were involved in the identification of two of the Gpr126 ligands (laminin-211, PrP-C). Without such testing, the invention of a ligand-binding role for the calcium binding site is premature, confusing and thus misleading. Thus, the authors should provide evidence that abolishment of the calcium binding mode of the Gpr126 ECR is reflected changes in ligand binding, and signaling.

→ Although several other CUB domains have been shown to use their calcium-binding sites to bind to ligands, we agree that the statement that the Gpr126 calcium-binding site might also act as a ligand-binding site is speculative and we have not shown evidence to support this.

We have tried to reproduce the results that show that type IV collagen and PrP^C bind to Gpr126, as well as show a difference in ligand-binding between the -ss and +ss isoforms, but our results were inconclusive. Because of the lack of evidence for the calcium-binding site as a ligand-binding site, we have modified the manuscript and the model in Figure 7 to remove claims that are not supported by evidence.

The experiments we performed to test ligand-binding and signaling are described below. The ligands we tested were:

- 1) type IV collagen (Sigma-Aldrich, C5533) – This is the same catalog number as the one provided in (Petersen et al., 2015). In (Paavola et al., 2014), type IV collagen was obtained from Sigma-Aldrich but no catalog number was provided.
- 2) PrP^C FT (flexible tail; residues 23-34) – This peptide was synthesized by GenScript and has the same sequence (plus a conjugated biotin) as the peptide used in (Kuffer et al., 2016) which was synthesized by EZ Biosciences.
- 3) PrP^C 23-50 (residues 23-50) – This peptide was synthesized by GenScript and has the same sequence (plus a conjugated biotin) as the peptide used in (Kuffer et al., 2016) which was synthesized by EZ Biosciences.

Laminin-211 was not tested because (Petersen et al., 2015) found that laminin-211 binds to the more C-terminal portion of the ECR, which does not include the more N-terminal calcium-binding site on the CUB domain.

First, we tested binding of PrP^C FT to Gpr126/GPR126 using a pull-down assay, similar to that described in (Paavola et al., 2014). The results are shown in Figure X4, and the experimental details are described in the figure legend.

Figure X4. PrP^C FT binding tests on Gpr126/GPR126 splice isoforms.

(A) Binding tests of PrP^C FT to Gpr126/GPR126 using a pull-down assay. Biotinylated PrP^C FT (15uM, 30uL) was incubated with streptavidin beads for 15 minutes at 4C. Free streptavidin sites were then blocked with biotin for 15 minutes at 4C. The coated beads were washed twice with PBS, then incubated with purified hGPR126 +ss ECR R468A (15uM, 30uL), zfGpr126 -ss ECR (15uM, 30uL), zfGpr126 -ss D134A/F135A ECR (2uM, 30uL) overnight at 4C. The hGPR126 proteins were mutated to be furin-cleavage resistant (R468A) in order to avoid additional complexity to the experiment. In addition, the purified proteins were also incubated with a second set of control “empty beads” that were not coated with PrP^C FT (but were blocked with biotin). After the overnight incubation, the supernatants were collected (unbound fraction).

The beads (bound fraction) were washed twice with PBS. Both unbound and bound fractions were subjected to SDS-PAGE for analysis. Bands for autoproteolyzed ECR fragments (N-terminal: red; C-terminal: blue) and streptavidin monomer/dimer (orange) are boxed.

(B) Binding tests of PrP^C FT to Gpr126/GPR126 using a pull-down assay, as described in (A), except the PrP^C FT beads were incubated with purified hGPR126 -ss ECR R468A (15uM, 30uL), hGPR126 +ss ECR R468A (15uM, 30uL), zfGpr126 -ss ECR (15uM, 30uL), zfGpr126 +ss ECR (2uM, 30uL) overnight at 4C. Both unbound and bound fractions were subjected to SDS-PAGE for analysis. Bands for autoproteolyzed ECR fragments (N-terminal: red; C-terminal: blue) and streptavidin monomer/dimer (orange) are boxed. Red asterisks denote faint but visible bands.

(C) Bound fractions from (B) were subjected to SDS-PAGE and western blotting (1°: mouse anti-His (Qiagen, 34660); 2°: donkey anti-mouse Alexa Fluor 488 (Invitrogen, A-21202)) for additional analysis.

From the gel in Figure X4A, we see that most of the protein are in the unbound fractions. The GAIN domain-cleaved ECRs are represented as two bands (red: N-terminal fragment; blue: C-terminal fragment). In the bound fraction, the streptavidin monomer and dimer proteins from the beads are present (orange) in addition to single bands that run at the same molecular weights as the zfGpr126 -ss, zfGPR126 -ss D134A/F135A, and hGPR126 +ss N-terminal fragment bands. This suggests that the purified ECR proteins are binding to the PrP^C FT-coated beads. However, the empty beads also show binding of at least 2 out of 3 of the proteins, indicating that any binding we have observed may be non-specific. And thus, we believe these experiments, at their current state, fail to demonstrate a reliable direct interaction of Gpr126/GPR126 with the tested ligands. We thus make no such claims.

We still performed additional experiments. If interested: Another pull-down assay testing both -ss and +ss isoforms from zebrafish and human showed most of the protein in the unbound fractions but some binding of all proteins to PrP^C FT-coated beads (Figure X4B). The bound fractions were also subjected to western blotting (Figure X4C) which showed that both -ss and +ss isoforms from zebrafish and human appear to bind to PrP^C FT-coated beads, though there does not appear to be a consistent difference between -ss and +ss isoforms. However, since we observed non-specific binding to empty beads earlier (Figure X4A), we cannot be sure that Gpr126/GPR126 binding to PrP^C FT is specific.

Figure X5. Type IV collagen binding tests on Gpr126/GPR126 splice isoforms.

(A) Binding tests of type IV collagen to Gpr126/GPR126 using a pull-down assay. Biotinylated type IV collagen (15uM, 30uL) was incubated with streptavidin beads for 15 minutes at 4C. Free streptavidin sites were then blocked with biotin for 15 minutes at 4C. The coated beads were washed twice with PBS, then incubated with purified hGPR126 -ss ECR R468A (15uM, 30uL), hGPR126 +ss ECR R468A (15uM, 30uL), zfGpr126 -ss ECR (15uM, 30uL), zfGpr126 +ss ECR (2uM, 30uL) overnight at 4C. The hGPR126 proteins were mutated to be furin-cleavage resistant (R468A) in order to avoid additional complexity to the experiment. After the overnight incubation, the supernatant was collected (unbound fraction). The beads (bound fraction) were washed twice with PBS. Both unbound and bound fractions were subjected to SDS-PAGE for analysis. Bands for autoproteolyzed ECR fragments (N-terminal: red; C-terminal: blue) and streptavidin monomer/dimer (orange) are boxed.

(B) Bound fractions were subjected to SDS-PAGE and western blotting (1°: mouse anti-His (Qiagen, 34660); 2°: donkey anti-mouse Alexa Fluor 488 (Invitrogen, A-21202)) for additional analysis.

We also performed the same pull-down assay testing for binding of type IV collagen to Gpr126/GPR126. In Figure X5A, we again see that most of the protein are in the unbound fractions. In the bound fractions, the streptavidin monomer and dimer proteins from the beads are present (orange) as well as many other bands. Because type IV collagen runs as many bands on the gel, it is difficult to differentiate potential bound Gpr126/GPR126 protein bands from the bead-linked type IV collagen bands. We purchased several of batches of type IV collagen from different companies and all seem problematic when we run on the gel.

To determine whether there are Gpr126/GPR126 bands hidden by the type IV collagen bands, we performed a western blot to detect only the His-tagged ECR proteins (Figure X5B). Similar to the PrP^C FT binding assay, both -ss and +ss isoforms from zfGpr126 and hGPR126 appear to bind to type IV collagen-coated beads, and there does not appear to be a consistent difference between -ss and +ss isoforms. However, since we observed non-specific binding of the proteins to empty beads previously (Figure X4A), we cannot rule out the possibility that the binding of Gpr126/GPR126 to type IV collagen is non-specific. Thus, we again make no claims.

→ In addition to ligand-binding experiments, we have also performed cAMP signaling assays (GloSensor cAMP assay from Promega). HEK293 cells expressing Gpr126/GPR126 are treated with type IV collagen and PrP^C. Similar experiments using cells transfected with Gpr126/GPR126 were done in previous studies:

(Petersen et al., 2015) – type IV collagen (2ug/mL) treatment on COS-7 cells expressing hGPR126 activates cAMP signaling using CREB-luciferase reporter gene assay

(Paavola et al., 2014) – type IV collagen (3ug/mL) treatment on HEK293 cells expressing zfGpr126 activates cAMP signaling using cAMP-ELISA (EMD Millipore)

(Kuffer et al., 2016) – PrP^C FT (0.5uM) treatment on HEK293T cells expressing zfGpr126 activates cAMP signaling using cAMP-ELISA (Enzo Life Sciences)

The results (Figure X6) show that in all cases, there were no significant differences between the PBS-treated control and the ligand-treated samples. As (Petersen et al., 2015) notes that type IV collagen treatment was only observed using a more sensitive CREB-luciferase reporter gene

assay and not the AlphaScreen cAMP assay kit (PerkinElmer Life Sciences), it is likely that our GloSensor cAMP assay is also not detecting possible ligand-mediated signaling effects.

In summary, we observed no effect of ligands on receptor signaling. We make no claims in the manuscript.

- If closed and open ECR conformations represent two signaling states of Gpr126, and if this transition can be influenced by calcium as a factor that stabilizes the closed conformation of the receptor, this process should be physiologically prone to changes in calcium level and likely

dynamic. These changes may occur either individually or simultaneously in the ER/Golgi/secretory pathway and/or in the extracellular space. Calcium levels can be easily manipulated in in vitro signaling assays as conducted by the authors by changing extracellular calcium levels through buffers or by changing the composition of the medium. Intracellular and intra-ER calcium levels can be manipulated through the use of SERCA inhibitors such as thapsigargin. This will allow to test whether calcium changes impinge on signaling outcome of the -ss and +ss forms, and should be abolished in the calcium binding site mutants of either receptor isoform substantiating the speculation that conformational transitions between closed and open ECR conformation are attributable for this effect. Note: I am not asking for further structural studies under different calcium conditions as I am aware of the substantial efforts required to conduct these assays. But at least one independent line of evidence should underpin the speculation that the calcium binding site identified in Gpr126 regulates signaling behavior of the receptor through its role in the ECR structure.

→ We thank the reviewer for this suggestion that would strengthen our argument that the calcium-binding site plays an important functional role in Gpr126. We have performed the suggested experiment of treating Gpr126-transfected cells with EGTA and thapsigargin using concentrations commonly used in the literature. However, we note that whereas more traditional assays using these reagents test for a response immediately after treatment, in our assay we are treating with EGTA/thapsigargin for several hours before measurement to allow time for a signaling response. Ultimately, our results are inconclusive due to non-specific effects of EGTA and thapsigargin on the cells, causing unreliable signal. And thus, we do not make any claims about calcium-dependent conformational changes. We also want to note that due to high and consistent levels of calcium in the extracellular space, calcium is usually not used as a regulatory molecule outside of the cell although it is often important for structural stability of protein complexes (such as neuroligin and neuroligin interaction).

The results of the experiment (treatment 45 hours post-transfection, 3 hours before signal readout) are shown in Figure X7, which show no significant differences between the PBS-treated and EGTA-treated samples, nor between the DMSO(0.1%)-treated and thapsigargin-treated samples.

	EV	zfGpr126 (-ss)	zfGpr126 (+ss)	β2AR
PBS vs. EGTA	ns	ns	ns	ns
DMSO vs. thapsigargin	ns	ns	ns	**

Figure X7. Effect of EGTA/thapsigargin on Gpr126 cAMP signaling.

(top, left) cAMP signaling levels for HEK293 cells transfected with empty vector (EV) and zebrafish Gpr126 splice isoforms, treated with either PBS or 2mM EGTA 45 hours post-transfection (3 hours before measurement).

(top, right) cAMP signaling levels for HEK293 cells transfected with empty vector (EV) and zebrafish Gpr126 splice isoforms, treated with either 0.1% DMSO or 1uM thapsigargin 45 hours post-transfection (3 hours before measurement).

(bottom) Analysis of statistical significance: ns, $P > 0.05$; *, $P \leq 0.05$; **, $P \leq 0.01$; ***, $P \leq 0.001$; ****, $P \leq 0.0001$; by two-way ANOVA and Tukey's multiple comparisons test.

Because we did not see a difference, we also tried an earlier treatment (21 hours post-transfection, 27 hours before signal readout) (Figure X8). In this experiment, we see significantly lower cAMP signal for both zfGpr126 -ss and +ss between PBS-treated and EGTA-treated samples as well as between DMSO-treated and thapsigargin-treated samples. This might suggest that calcium plays a role in the regulation of Gpr126 signaling. However, because we also see significant changes in signaling between PBS-treated and EGTA-treated β2AR as well as DMSO-treated and thapsigargin-treated β2AR, it is likely that the EGTA and thapsigargin are affecting the cells in a non-specific manner since β2AR is not known to bind to calcium or mediate calcium-dependent signaling. In addition, we observed that the cells treated with either EGTA or thapsigargin looked unhealthy (rounder, semi-detached) compared to the seemingly healthy cells that were treated with PBS or DMSO. We think that the EGTA and thapsigargin treatments are toxic to the cells and this may contribute to the observed lower cAMP levels. Altogether, we conclude that any effects of EGTA or thapsigargin on Gpr126 signaling cannot be definitively due to removal of calcium from the calcium-binding sites of Gpr126 receptors. And thus, we make no claims about the effect of calcium-dependent conformational changes.

We have modified the manuscript and removed our claims and changed the model in Figure 7. We think this will address the reviewer's concerns.

Figure X8. Effect of EGTA/thapsigargin on Gpr126 cAMP signaling.

(top, left) cAMP signaling levels for HEK293 cells transfected with empty vector (EV) and zebrafish Gpr126 splice isoforms, treated with either PBS or 2mM EGTA 21 hours post-transfection (27 hours before measurement).

(top, right) cAMP signaling levels for HEK293 cells transfected with empty vector (EV) and zebrafish Gpr126 splice isoforms, treated with either 0.1% DMSO or 1uM thapsigargin 21 hours post-transfection (27 hours before measurement).

(bottom) Analysis of statistical significance: ns, $P > 0.05$; *, $P \leq 0.05$; **, $P \leq 0.01$; ***, $P \leq 0.001$; ****, $P \leq 0.0001$; by two-way ANOVA and Tukey's multiple comparisons test.

- The authors should mention and discuss why the tethered agonist peptide used in their Gpr126 stimulation assay differs significantly in size from the previously published one in Liebscher et al., 2014. The authors here used a p7 peptide, while in the original paper a p16 was identified as the most potent one from a library of gradually elongated tethered agonist portions and used to stimulate activity of Gpr126. This seems important as the p7 peptide reported in Liebscher et al. did not affect cAMP accumulation over control conditions. Have the authors tested a p16 peptide as well, what was the outcome compared to the p7 variant?

→ We thank the reviewer for bringing up this important point. We believe that there is a difference in peptide sensitivity between the zebrafish and human receptors, as well as differences in constructs used in our study compared to (Liebscher et al., 2014).

In (Liebscher et al., 2014), human GPR126 is used in the signaling assays. In the experiment where the authors tested peptides of increasing length on P2Y₁₂-ΔGPS-CTF (a truncated and chimeric GPR126 construct), addition of 1mM p7 increased cAMP levels ~2-fold over untreated (Figure X9A). Addition of 1mM of the longer p16 peptide increased cAMP levels over the untreated sample by ~8-fold (Figure X9A). They also show that, using wild-type GPR126, treatment with 1mM p16 increased cAMP levels ~10-fold (Figure X9B).

(Liebscher et al., 2014)

Figure X9. GPR126 Agonistic Peptides Are Derived from the C-Terminal Part of the GPS.

(A) Application of 1 mM peptides of different lengths derived from the C-terminal part of the GPS, beginning at the cleavage site of GPR126, revealed agonistic properties as measured by cAMP accumulation. The highest agonistic efficacy was detected for a peptide containing 16 aa (p16). Negative controls: eV and GPR126-P2Y₁₂-DGPS-CTF mutant. Basal cAMP levels were 3.8 ± 1.6 nM.

(B) Different p16 concentrations were tested on WT P2Y₁₂, WT GPR126, and P2Y₁₂-DGPS-CTF. Inset: the concentration-response curve of p16 at WT GPR126 revealed an EC₅₀ > 400 nM. Basal eV levels were 3.2 ± 0.7 nM.

In our signaling assay (Figure X10), we see that both isoforms of zebrafish Gpr126 are activated by both p7 and p14 peptides. Zebrafish Gpr126 is activated only slightly more by 1mM p14 compared to 100uM p7, which is why we chose to use 100uM p7 in our activation assay. However, human GPR126 is only activated by p14 (~10-fold) and not p7. This could be due to an inherent difference between zebrafish Gpr126 and human GPR126 transmembrane domains which respond differently to different lengths of peptide. However, since (Liebscher et al., 2014) showed activation of the truncated P2Y₁₂-ΔGPS-CTF GPR126 construct by shorter peptides, another possibility is that the human GPR126 transmembrane domain is blocked by its ECR, more so than the zebrafish version. This would prevent access of the p7 peptide on the full-length human GPR126. In this case, activation would be observed only when the more potent p14/p16 peptide is applied.

We have modified Supplementary Figure 4C to show activation by p14 instead of p7 and have included the data for human GPR126 constructs as well.

Figure X10. Effect of p7 and p14 synthetic peptide on Gpr126/GPR126 cAMP signaling. cAMP basal signaling levels for HEK293 cells transfected with empty vector (EV), zebrafish Gpr126 splice isoforms, and human GPR126 splice isoforms, treated with 2% DMSO, 100uM p7, or 1mM p14 15 minutes before measurement.

p16, p390-392

The statement with regards to ligand binding ('the importance of furin cleavage is likely not primarily important for proper expression and trafficking but rather for a ligand-mediated function') has no experimental foundation. I would request to remove the statement regarding a potential ligand interaction through the SEA domain, or ask the authors to demonstrate evidence for ligand binding at the SEA domain in relation to furin processing (remove furin activity, remove furin cleavage site).

→ We agree that this is speculative and have removed the last part of this sentence.

Fig. S6

The authors mention that they engineered a furin-cleavage resistant Gpr126 mutant but show no evidence for this effect on furin cleavage. The authors are asked to demonstrate cleavage-deficiency effected by the mutation, e.g. by a Western blot.

I assume that R468A represents this mutation? If so, this is neither mentioned in the main text nor the figure caption, but only in the diagrams in S6B-D. Insert a suitable explanation in main text and caption.

→ R468A indeed represents the furin-cleavage mutant. We have included protein gels in Supplementary Figure 6 that show that ~50% of wild-type human GPR126 -ss and +ss isoforms are cleaved and that the R468A mutants completely abolish furin cleavage in both -ss and +ss isoforms. We also note that this mutation was first shown to be furin-cleavage resistant in a previous study (Moriguchi et al., 2004) and have clarified this in the manuscript. In addition, we

have removed Supplementary Figures 5C-D due to unreliable results but have provided explanations of S5B (now S5E) in the main text and caption.

MINOR POINTS

p3, 67-68

There are only 32 human aGPCR genes currently known, EMR4 is considered a pseudogene in the human genome. Please correct.

→ We have corrected this number throughout the manuscript.

p5, 120-121

'... 150 aa region between PTX and HormR, could not be identified.'

I'd suggest to rephrase since apparently the fact that there is a linker region between the said domains, to which a structure could not be readily assigned, was already known.

'... 150 aa region between PTX and HormR, could not be assigned to a known structural fold.'

→ We have changed this sentence accordingly. However, we note that the presence of the linker region was not known prior to the current study.

Figure S4E

A grouping of observed conformations such as presented in Fig. 3D is missing for this dataset. Please provide.

→ We have now included the grouped class averages (now Supplementary Figure 4D).

Figure caption S4C

'Basal signaling for sam constructs ...'. Correct

→ We have corrected this typo.

p15, 370

Insert bold words in statement: 'Although THE furing cleavage SITE is not conserved..... '

→ We have modified this sentence accordingly.

p16, 388-389

There is a problem with grammar of this sentence:

'HEK293 cells transfected with a mutant form of human GPR126 that is not able to be cleaved by furin were detected on the cell surface. '

Change to:

'Human GPR126 that is not able to be cleaved by furin was transfected in HEK293 cells and detected on the cell surface. '

→ We have modified this sentence accordingly.

Fig. 7E

Is the ECR shed after furin cleavage? There is not evidence for this in this manuscript, and this would also be in contrast to data obtained from the Notch receptor field showing that furin cleavage results in a non-covalent heterodimer (similar the SEA and GAIN domain

autoproteolysis), which does not readily dissociate (see Logeat et al., PNAS, 1998; Blaumueller et al., Cell 1997, Rand et al., Mol Cell Biol, 2000).

Hence, the cartoon introduces a situation - NTF shedding - that is has not been formally experimentally associated with neither furin cleavage, SEA domain autoproteolysis nor GAIN domain self-cleavage. The latter point has been only partially addressed by artificially or genetically produced Gpr126 NTF, which can rescue Gpr126 hypomorphic phenotypes in vivo. However, to my knowledge it was not tested whether these phenotypes in heart development can be complemented by using a GAIN domain autoproteolysis-disrupting mutation in Gpr126. In sum, the scenario of NTF shedding is very speculative and further elaboration on this without concrete evidence is misleading readers.

→ We agree that GPR126 furin cleavage probably acts similarly to cleaved SEA domains from Notch in that cleavage does not immediately lead to shedding, and this is supported by Supplementary Figure 6B showing elution of furin-cleaved ECR fragments together from a size exclusion column at the same volume as the full-length uncleaved ECR. However, we agree that the shedding model is speculative, and we have modified this figure to simplify it and to only show scenarios for which we have provided evidence.

Kuffer, A., Lakkaraju, A.K., Mogha, A., Petersen, S.C., Airich, K., Doucerain, C., Marpakwar, R., Bakirci, P., Senatore, A., Monnard, A., *et al.* (2016). The prion protein is an agonistic ligand of the G protein-coupled receptor Adgrg6. *Nature* 536, 464-468.

Liebscher, I., Schon, J., Petersen, S.C., Fischer, L., Auerbach, N., Demberg, L.M., Mogha, A., Coster, M., Simon, K.U., Rothmund, S., *et al.* (2014). A tethered agonist within the ectodomain activates the adhesion G protein-coupled receptors GPR126 and GPR133. *Cell reports* 9, 2018-2026.

Moriguchi, T., Haraguchi, K., Ueda, N., Okada, M., Furuya, T., and Akiyama, T. (2004). DREG, a developmentally regulated G protein-coupled receptor containing two conserved proteolytic cleavage sites. *Genes to cells : devoted to molecular & cellular mechanisms* 9, 549-560.

Paavola, K.J., Sidik, H., Zuchero, J.B., Eckart, M., and Talbot, W.S. (2014). Type IV collagen is an activating ligand for the adhesion G protein-coupled receptor GPR126. *Science signaling* 7, ra76.

Petersen, S.C., Luo, R., Liebscher, I., Giera, S., Jeong, S.J., Mogha, A., Ghidinelli, M., Feltri, M.L., Schoneberg, T., Piao, X., *et al.* (2015). The adhesion GPCR GPR126 has distinct, domain-dependent functions in Schwann cell development mediated by interaction with laminin-211. *Neuron* 85, 755-769.

Reviewers' comments:

Reviewer #1 (Remarks to the Author):

The authors have dealt with all the comments in my review satisfactorily.

Reviewer #2 (Remarks to the Author):

The authors have answered all of my concerns.

Reviewer #3 (Remarks to the Author):

The authors have made the necessary changes and text edits and I am satisfied with the state of the paper. This is a good solid piece of work.

Reviewer #4 (Remarks to the Author):

Please see next page.

Remarks to the authors

I thank the authors for the meticulous revision of their paper, which has substantially improved. Regarding the rectification of uncertainties pertaining to the interpretation of datasets I am satisfied. In order to keep track of the extensive changes that followed upon my inquiries I listed all points that the authors dropped since their initial claims could not be confirmed after further experimentation, which I like to list here again also for the consideration of the other referees and the editor:

Re Calcium-binding site/structural changes/signaling interconnection

- 'We are aware that the simple model contradicts the data; and that the more complex and ill-explained model agrees with the data. For this reason, we prefer to move the mutant data to Supplementary Figure 4H and remove claims about it and do not mention the complex model.'

Re Ligand-binding at the calcium binding site:

- 'Because of the lack of evidence for the calcium-binding site as a ligand-binding site, we have modified the manuscript and the model in Figure 7 to remove claims that are not supported by evidence.'
- 'We have tried to reproduce the results that show that type IV collagen and PrPC bind to Gpr126, as well as show a difference in ligand-binding between the -ss and +ss isoforms, but our results were inconclusive.'
- 'We observed no effect of ligands on receptor signaling. We make no claims in the manuscript.'

Re Calcium-dependent conformational changes:

- 'We do not make any claims about calcium-dependent conformational changes.'

Re Furin cleavage and signaling:

- Removal of Supplementary Figures 6C-D due to unreliable results

I like to repeat that I think that structural and in vivo data presented here are reliable. However, I would also like to point out that pharmacological data do not yet connect these concepts coherently. After the revision it seems clear that, unsurprisingly, only splicing affects signaling outcome of Gpr126 as a means of transcriptional control of Gpr126 function (see here for publications that have demonstrated this recently using other adhesion GPCRs: [10.1016/j.celrep.2019.01.040](https://doi.org/10.1016/j.celrep.2019.01.040), [10.1038/s41598-019-46265-x](https://doi.org/10.1038/s41598-019-46265-x)), but neither calcium, calcium binding, or ligand binding to the calcium binding site as post-translational means do.

This regretful situation notwithstanding, I believe that the novel structure of the complete ECR of Gpr126 is interesting and deserves publication in Nat Commun.

I would ask that a few remarks, largely pertaining to negative data that accumulated throughout the revision process, be mentioned in the main text in order to avoid the confusion for the reader that I faced when first reading the manuscript. Comments on this are inserted below next to the responses of the authors. I do not need to see the final revision of the manuscript again if the editorial office can ensure that the points were included in the manuscript.

Response to Reviewer Comments

Structural basis for adhesion G protein-coupled receptor Gpr126 function
NCOMMS-19-13915-T

Reviewer #4 (Remarks to the Author):

The authors present a set of structural, pharmacological and in vivo assays using zebrafish and uncover novel aspects of the adhesion GPCR Gpr126. The authors are experts in the analysis of this receptor class and the manuscript contains the description of several novel, insightful features of this interesting receptor group. Foremost, the identification of hitherto unknown SEA domain in the ECR of Gpr126 is interesting, and the structure-based evidence on different ECR conformations that may result from alternative splicing and calcium binding are of high and general interest. Unfortunately, the manuscript is partly confusing and imprecise when it comes to the integration of structural and pharmacological datasets, and completely lacks evidence for speculations raised on the effects of the identified structural properties of the ECR and ligand engagement. The authors need to provide further evidence in order to substantiate their claims how structure impacts on the function of the receptor in the context of ligand and calcium binding. In the current form I cannot recommend publication of the manuscript, but I have laid out pivotal experiments that I would like to see in order to underpin and bridge the solid structural and physiological datasets with sound pharmacological evidence.

→ We thank the reviewer for his/her constructive comments and for suggesting a path to improve the manuscript. We agree with the reviewer and understand his/her concerns. We believe we addressed all concerns and changed the manuscript accordingly. We present our answers/new results and other discussion below. We apologize for the lengthy discussion below. In summary:

-We performed ligand binding experiments and signaling assays with PrP^C and type IV collagen and were unable to detect reliable binding or effects on receptor signaling.

-We performed the suggested experiments using EGTA and thapsigargin and obtained unreliable results.

-Thus, we shuffled our calcium-binding mutant data to a supplementary figure rather than a main figure and changed the model in Figure 7 and have removed our claims about the calcium-binding site being a ligand-binding site.

MAJOR POINTS

p8, 177

The hLPHN3 GAIN structure was not published in Arac et al., 2012 so the reference is misplaced and should be moved before hLphn3 in the sentence.

Further, I was astounded to find the hLPHN3 GAIN domain displayed in this manuscript uncommented. There is no explanation as to whether the rendering displayed in Fig. S1F is based on a homology model (if so, a short section in the methods section on how was this done is required) or a novel X-ray crystallographic structure (if so, a PDB ID, the entire experimental description of its production and solution and short discussion in the text is required).

→ The hLPHN3 GAIN structure is a novel X-ray crystallographic structure from our lab that is

currently unpublished. We have removed it from the figure to avoid confusion

Figures 4A-C, S4B-D, S6B-D

Error bars are missing in the diagrams in 4A, S4B and S6B. Please provide those along with a statistical measure testing whether the displayed groups differ in surface expression levels from each other.

4 We acknowledge the lack of error bars in these figures and hope this will offer clarification: The data in the referenced figures show one signaling assay that is representative of at least three separate experiments. Each experiment includes a signaling assay and a cell-surface expression assay performed on the same transfected population of cells to ensure cell-surface expression level that directly matches to the cells that were used in the particular signaling assay. Our method for measuring cell-surface expression is to stain transfected cells (in parallel with cAMP measurement) with antibodies (1°: mouse M2 anti-FLAG, 2°: donkey anti-mouse Alexa Fluor 488) and measure the median fluorescence intensity of the cells using flow cytometry. Because we measure fluorescence intensity values for numerous cells (~10,000) we take the median fluorescence intensity as the "cell-surface expression" value that is shown in the figures. Therefore for each experiment we are using a single, median expression value for each population of transfected cells and do not have error bars for these values. However, please see Figure X3 for expression data averaged among triplicate experiments, showing consistent expression levels for all constructs. We have also included raw flow cytometry data in Supplementary Figure 4A-B

OK. Please include a brief version of explanation in the caption as to why no error bars are shown for the surface expression value in Fig. 4A, S4B and S6B, i.e. the fact that shown is only a single flow cytometry measurement. I am still uncertain why the authors do not show the pooled instead of the current representative graph of all the three separate replicates they conducted as they indicate in the rebuttal letter. From what is apparent in Fig. X3 this should not affect the general outcome of this assay but would offer the reader a more generalizable view of this dataset.

p12

The authors provide compelling structural evidence for the importance of the calcium binding site in Gpr126 through mutagenesis approaches, which remove the calcium binding site. These datasets are very interesting and central to establish structure-function correlates and ultimately causality between the observed features of pr126. However, I feel that the authors need to focus more diligently on the interconnection and presentation of the many facets they have uncovered in this manuscript, and correct or further substantiate some of the claims they make here:

→ We thank the reviewer for the positive comments on the calcium-binding site work and constructive comments that will help strengthen the manuscript.

- This set of experiments shows that the calcium binding site bound to calcium stabilizes the closed conformation that is commonly observed in the -ss form, because mutations to the site result in an opening of the conformation adopting a +ss-like shape. In a second line of experiments the authors first establish that the -ss and +ss have different signaling effects that can be summarized as follows: -ss/closed conformation/reduced cAMP signal as opposed to +ss/open conformation/elevated cAMP signal.

→ We agree with these interpretations of our work. We have now rearranged Figure 4 and Supplementary Figure 4. We now show in the main figure that alternative splicing has an effect on signaling. We moved the calcium-binding mutation data to the supplement.

- Here, the logic as to how calcium binding factors into this framework, is ill-explained or ill-founded. The authors show that removal of the calcium binding site in the -ss form results in an open conformation as expected if the site stabilized the closed conformation of the ECR. However, no change in cAMP signaling was observed.

→ That is right.

In the contrary, the already opened +ss conformation suffered through the calcium site mutations in signaling strength.

→ That is right, too.

If the calcium binding site stabilizes the low-signaling -ss closed conformation of the receptor, then mutations of the site should have a stimulating effect on the -ss form and no effect on the high-signaling open +ss form in signaling assays.

→ This is the most reasonable and simplest model. However, the data does not fit to this model.

But the authors document the inverse situation.

→ That is right. We originally thought and still think that this data can be explained by a more complex (and yet ill-explained) model which is: In addition to the above simple model, activation also needs binding of a ligand to the calcium-binding site and that the ligand is always present in our experimental system (HEK cell cultures). In other words, the protein should be “open” and also “be able to bind to its ligand” simultaneously so that it can be active. Only the wild type +ss isoform meets these requirements, and thus only that protein is “more active” and the others are not.

Therefore, the sentence 'These results suggest that the calcium-binding site is important for increased signaling in the (+ss) isoform and is likely a functional site for Gpr126.' (p13, 311-313) is true but does not reflect or is even opposing the findings on the conformational role of the calcium binding site and the impact of conformational changes on receptor signaling such as proposed by the authors above.

→ We are aware that the simple model contradicts the data; and that the more complex and ill-explained model agrees with the data. For this reason, we prefer to move the mutant data to Supplementary Figure 4H and remove claims about it and do not mention the complex model. We believe this will address the reviewer's concern.

If interested: here is a more detailed explanation of what we think:

We agree that these sets of signaling data are confusing and seem to be contradictory to our argument that the calcium-binding site is important for cAMP signaling. We hope that the following explanation will provide better context for this argument. With regard to the observation that cells transfected with the +ss isoform have higher levels of cAMP compared to cells transfected with the -ss isoform, we believe this could be due to:

- 1) The change in ECR conformation from closed to more open results in differences in signaling, possibly by changing ECR-7TM regulatory interactions. This change is inherent to the ECR conformation itself and does not require additional factors.
- 2) The change in ECR conformation from closed to more open exposes the calcium-binding site which might act as a ligand-binding site and activate cAMP signaling.

If the first model is correct, we would expect to see an increase in signaling when the -ss isoform is mutated in the calcium-binding site, because the calcium-binding site contributes to a stable closed conformation. We would not expect a large signaling change when the +ss isoform is mutated in the calcium-binding site, because the calcium-binding site has less of a contribution to maintaining a closed conformation for this isoform, which is more open.

If the second model is correct, we would not expect to see a significant change in signaling when the -ss isoform is mutated in the calcium-binding site, since in both cases: a) wild-type -ss isoform and b) D134A/F135A calcium-binding site mutant of -ss isoform, the calcium-binding

site is a) obscured or b) mutated. Hence, we believe the calcium-binding site would be “inactive” in both scenarios and would not signal to a higher level. We would, however, expect a decrease in signaling upon mutation of the calcium-binding site in the +ss isoform. The wildtype +ss, the ECR conformation is open and the calcium-binding site would be exposed to ligands in the extracellular matrix, which would lead to an increased signal (compared to the -ss isoform). Mutation of the calcium binding site in the +ss isoform would not affect the ECR conformation but it would disrupt the calcium-binding site such that ligands would not be able to bind there and activate signaling.

The signaling assay results suggest that the change in ECR conformation alone is not the reason for the higher signaling observed in the +ss isoform. We think that the role of the calcium-binding site is to act as a binding interface for at least two regulatory modes. The first, when the ECR conformation is closed, the calcium-binding site is used to bind to a distant site on Gpr126. This obscures the calcium-binding site from ligand binding events and keeps basal signaling at a lower level. The second, when the ECR conformation is open, the calcium-binding site is exposed to endogenous ligands and signaling can be activated to a higher level.

This is our explanation of the data. However, we did not include these claims in the manuscript.

I thank to authors for reassessing this part of the manuscript and toning down their claims drawn from the collective assessment of structural and pharmacological experiments in this paper. However, in the main manuscript it still reads:

'These results suggest that the calcium-binding site is important for increased signaling in the (+ss) isoform and may be a functional site for Gpr126.'

It is not made clear in the text and even more confusing to the reader that structural, functional and calcium-binding site mutational data are at a mismatch with each other here, especially after moving the mutation dataset out of the direct view of readers to the supplement. If the authors 'agree that these sets of signaling data are confusing and seem to be contradictory to our argument that the calcium-binding site is important for cAMP signaling'.

Please say so at this point in the main manuscript in the summary sentence of this results paragraph.

At the moment, there is a logic connection insinuated by the authors that cannot be substantiated. I see no flaw in actively making the readers aware of this disconnect, but argue that this makes the main messages of the manuscript even more clearer, credible and stimulating. So please comment on this in the results and discussion.

The authors have delivered a meticulous discussion of alternative explanations in the rebuttal letter, which I suggest to condense and transfer to the manuscript's discussion on this matter.

- Further, there is no mention and no experimental evidence providing an explanation as to why the calcium binding site may have a third function, additional to the closed conformation

stabilization and cAMP signal suppression: as a binding interface to ligands. Only on p17 (414-417) the authors introduce this concept in the discussion: 'Thus, we speculate that Gpr126 uses this site to not only bind to itself in a closed conformation (-ss isoform), effectively hiding the calcium-binding site from ligands, but also to bind to ligands when the ECR is in a possible open conformation (+ss isoform) in order to activate signaling.'

As the manuscript does not contain any experimental effort to test for differing ligand interactions between the -ss and +ss isoforms, nor the effect of abolishing the calcium binding site on ligand engagement and signaling, this statement is a clear overinterpretation of the data that are currently provided. Gpr126 has three identified ligands and none was tested in this manuscript. This could have been performed by the authors given their superb protein biochemical and genetic expertise, and backed by the fact that some of the co-authors were involved in the identification of two of the Gpr126 ligands (laminin-211, PrP^C). Without such testing, the invention of a ligand-binding role for the calcium binding site is premature, confusing and thus misleading. Thus, the authors should provide evidence that abolishment of the calcium binding mode of the Gpr126 ECR is reflected changes in ligand binding, and signaling.

→ Although several other CUB domains have been shown to use their calcium-binding sites to bind to ligands, we agree that the statement that the Gpr126 calcium-binding site might also act as a ligand-binding site is speculative and we have not shown evidence to support this.

We have tried to reproduce the results that show that type IV collagen and PrP^C bind to Gpr126, as well as show a difference in ligand-binding between the -ss and +ss isoforms, but our results were inconclusive. Because of the lack of evidence for the calcium-binding site as a ligand-binding site, we have modified the manuscript and the model in Figure 7 to remove claims that are not supported by evidence.

The experiments we performed to test ligand-binding and signaling are described below. The ligands we tested were:

- 1) type IV collagen (Sigma-Aldrich, C5533) – This is the same catalog number as the one provided in (Petersen et al., 2015). In (Paavola et al., 2014), type IV collagen was obtained from Sigma-Aldrich but no catalog number was provided.
- 2) PrP^C FT (flexible tail; residues 23-34) – This peptide was synthesized by GenScript and has the same sequence (plus a conjugated biotin) as the peptide used in (Kuffer et al., 2016) which was synthesized by EZ Biosciences.
- 3) PrP^C 23-50 (residues 23-50) – This peptide was synthesized by GenScript and has the same sequence (plus a conjugated biotin) as the peptide used in (Kuffer et al., 2016) which was synthesized by EZ Biosciences.

Laminin-211 was not tested because (Petersen et al., 2015) found that laminin-211 binds to the more C-terminal portion of the ECR, which does not include the more N-terminal calcium-binding site on the CUB domain.

First, we tested binding of PrP^C FT to Gpr126/GPR126 using a pull-down assay, similar to that described in (Paavola et al., 2014). The results are shown in Figure X4, and the experimental details are described in the figure legend.

Figure X4. PrP^C FT binding tests on Gpr126/GPR126 splice isoforms.

(A) Binding tests of PrP^C FT to Gpr126/GPR126 using a pull-down assay. Biotinylated PrP^C FT (15uM 30uL) was incubated with streptavidin beads for 15 minutes at 4°C. Free streptavidin sites were then blocked with biotin for 15 minutes at 4°C. The coated beads were washed twice with PBS then incubated with purified hGPR126 +ss ECR R468A (15uM 30uL), zfGpr126 -ss ECR (15uM, 30uL) zfGpr126 -ss D134A/F135A ECR (2uM, 30uL) overnight at 4°C. The hGPR126 proteins were mutated to be furin-cleavage resistant (R468A) in order to avoid additional complexity to the experiment. In addition, the purified proteins were also incubated with a second set of control "empty beads" that were not coated with PrP^C FT (but were blocked with biotin). After the overnight incubation, the supernatants were collected (unbound fraction). The beads (bound fraction) were washed twice with PBS. Both unbound and bound fractions were subjected to SDS-PAGE for analysis. Bands for autoproteolyzed ECR fragments (N-terminal: red; C-terminal: blue) and streptavidin monomer/dimer (orange) are boxed.

(B) Binding tests of PrP^C FT to Gpr126/GPR126 using a pull-down assay, as described in (A), except the PrP^C FT beads were incubated with purified hGPR126 -ss ECR R468A (15uM, 30uL) hGPR126 +ss ECR R468A (15uM 30uL), zfGpr126 -ss ECR (15uM, 30uL), zfGpr126 +ss ECR (2uM, 30uL) overnight at 4°C. Both unbound and bound fractions were subjected to SDS-PAGE for analysis. Bands for autoproteolyzed ECR fragments (N-terminal: red; C-terminal: blue) and streptavidin monomer/dimer (orange) are boxed. Red asterisks denote faint but visible bands.

(C) Bound fractions from (B) were subjected to SDS-PAGE and western blotting (1: mouse anti-His (Qiagen, 34660); 2: donkey anti-mouse Alexa Fluor 488 (Invitrogen A-21202)) for additional analysis.

From the gel in Figure X4A, we see that most of the protein is in the unbound fractions. The GAIN domain-cleaved ECRs are represented as two bands (red: N-terminal fragment; blue: C-terminal fragment). In the bound fraction the streptavidin monomer and dimer proteins from the beads are present (orange) in addition to single bands that run at the same molecular weights as the zfGpr126 -ss, zfGPR126 -ss D134A/F135A, and hGPR126 +ss N-terminal fragment bands. This suggests that the purified ECR proteins are binding to the PrP^C FT-coated beads. However, the empty beads also show binding of at least 2 out of 3 of the proteins indicating that any binding we have observed may be non-specific. And thus, we believe these experiments, in their current state fail to demonstrate a reliable direct interaction of Gpr126/GPR126 with the tested ligands. We thus make no such claims.

We still performed additional experiments. If interested: Another pull-down assay testing both -ss and +ss isoforms from zebrafish and human showed most of the protein in the unbound fractions but some binding of all proteins to PrP^C FT-coated beads (Figure X4B). The bound fractions were also subjected to western blotting (Figure X4C) which showed that both -ss and

+ss isoforms from zebrafish and human appear to bind to PrP^C FT-conted beads though there does not appear to be a consistent difference between -ss and +ss isoforms. However, since we observed non-specific binding to empty beads earlier (Figure X4A), we cannot be sure that Gpr126/GPR126 binding to PrP^C FT is specific.

Figure XS. Type IV collagen binding tests on Gpr126/GPR126 splice isoforms. (A) Binding tests of type IV collagen to Gpr126/GPR126 using a pull-down assay. Biotinylated type IV collagen (15uM, 30uL) was incubated with streptavidin beads for 15 minutes at 4C. Free streptavidin sites were then blocked with biotin for 15 minutes at 4C. The coated beads were washed twice with PBS then incubated with purified hGPR126 ss ECR R468A (15uM, 30uL), hGPR126 +ss ECR R468A (15uM, 30uL), zfGpr126 -ss ECR (15uM, 30uL), zfGpr126 +ss ECR (2uM, 30uL) overnight at 4C. The hGPR126 proteins were furin-cleavage resistant (R468A) in order to avoid additional complexity to the experiment. After the overnight incubation the supernatant was collected (unbound fraction). The beads (bound fraction) were washed twice with PBS. Both unbound and bound fractions were subjected to SDS-PAGE for analysis. Bands for autoproteolyzed ECR fragments (N-terminal: red; C-terminal: blue) and streptavidin monomer/dimer (orange) are boxed. (B) Bound fractions were subjected to SDS-PAGE and western blotting (1°: mouse anti-His (Qingen 34660); 2°: donkey anti-mouse Alexa Fluor 488 (Invitrogen, A-21202)) for additional analysis.

We also performed the same pull-down assay testing for binding of type IV collagen to Gpr126/GPR126. In Figure XSA, we again see that most of the protein are in the unbound fractions. In the bound fractions, the streptavidin monomer and dimer proteins from the beads are present (orange) as well as many other bands. Because type IV collagen runs as many bands on the gel, it is difficult to differentiate potential bound Gpr126/GPR126 protein bands from the background type IV collagen bands. We purchased several of batches of type IV collagen from different companies and all seem problematic when we run on the gel.

To determine whether there are Gpr126/GPR126 bands hidden by the type IV collagen bands, we performed a western blot to detect only the His-tagged ECR proteins (Figure XSB). Similar to the PrP^C FT binding assay, both -ss and +ss isoforms from zfGpr126 and hGPR126 appear to bind to type IV collagen-coated beads, and there does not appear to be a consistent difference between -ss and +ss isoforms. However, since we observed non-specific binding of the proteins

to empty beads previously (Figure X4A), we cannot rule out the possibility that the binding of Gpr126/GPR126 to type IV collagen is non-specific. Thus, we again make no claims.

→ In addition to ligand-binding experiments, we have also performed cAMP signaling assays (GloSensor cAMP assay from Promega). HEK293 cells expressing Gpr126/GPR126 are treated with type IV collagen and PrP^C. Similar experiments using cells transfected with Gpr126/GPR126 were done in previous studies:

(Petersen et al., 2015) – type IV collagen (2ug/mL) treatment on COS-7 cells expressing hGPR126 activates cAMP signaling using CREB-luciferase reporter gene assay

(Paavola et al., 2014) – type IV collagen (3ug/mL) treatment on HEK293 cells expressing zfGpr126 activates cAMP signaling using cAMP-ELISA (EMD Millipore)

(Kuffer et al., 2016) – PrP^C FT (0.5uM) treatment on HEK293T cells expressing zfGpr126 activates cAMP signaling using cAMP-ELISA (Enzo Life Sciences)

The results (Figure X6) show that in all cases, there were no significant differences between the PBS-treated control and the ligand-treated samples. As (Petersen et al., 2015) notes that type IV collagen treatment was only observed using a more sensitive CREB-luciferase reporter gene assay and not the AlphaScreen cAMP assay kit (PerkinElmer Life Sciences), it is likely that our GloSensor cAMP assay is also not detecting possible ligand-mediated signaling effects.

Figure X6. Effect of type IV collagen and PrP^C on Gpr126/GPR126 cAMP signaling.
 (A) cAMP signaling levels for HEK293 cells transfected with empty vector (EV) and human GPR126 splice isoforms, treated with either PBS or 10ug/mL type IV collagen two hours before measurement.
 (B) Same as (A) but treated with 10uM PrP²³⁻⁵⁰.
 (C) Same as (A) but treated with 10uM PrP^C FT.
 (D) cAMP signaling levels for HEK293 cells transfected with empty vector (EV) and zebrafish Gpr126 splice isoforms, treated with either PBS or 3ug/mL type IV collagen two hours before measurement.
 (E) Same as (D) but treated with 2uM PrP^C FT. ns, not significant.

In summary, we observed no effect of ligands on receptor signaling. We make no claims in the manuscript.

I thank for authors for their efforts and agree with the authors that the experiments on Gpr126 PrP-C-FT and collagen IV interactions 'at their current state, fail to demonstrate a reliable direct interaction of Gpr126/GPR126 with the tested ligands.' I also commend the authors on being frank about the failure to repeat these experiments that were previously highly visibly published (Paavola et al., Sci Signal 2014; Kuffer et al., Nature, 2016).

- If closed and open ECR conformations represent two signaling states of Gpr126, and if this transition can be influenced by calcium as a factor that stabilizes the closed conformation of the receptor, this process should be physiologically prone to changes in calcium level and likely dynamic. These changes may occur either individually or simultaneously in the ER/Golgi/secretory pathway and/or in the extracellular space. Calcium levels can be easily manipulated in in vitro signaling assays as conducted by the authors by changing extracellular calcium levels through buffers or by changing the composition of the medium. Intracellular and intra-ER calcium levels can be manipulated through the use of SERCA inhibitors such as thapsigargin. This will allow to test whether calcium changes impinge on signaling outcome of the -ss and +ss forms, and should be abolished in the calcium binding site mutants of either receptor isoform substantiating the speculation that conformational transitions between closed and open ECR conformation are attributable for this effect. Note: I am not asking for further structural studies under different calcium conditions as I am aware of the substantial efforts required to conduct these assays. But at least one independent line of evidence should underpin the speculation that the calcium binding site identified in Gpr126 regulates signaling behavior of the receptor through its role in the ECR structure.

→ We thank the reviewer for this suggestion that would strengthen our argument that the calcium-binding site plays an important functional role in Gpr126. We have performed the suggested experiment of treating Gpr126-transfected cells with EGTA and thapsigargin using concentrations commonly used in the literature. However, we note that whereas more traditional assays using these reagents test for a response immediately after treatment, in our assay we are treating with EGTA/thapsigargin for several hours before measurement to allow time for a signaling response. Ultimately, our results are inconclusive due to non-specific effects of EGTA and thapsigargin on the cells, causing unreliable signal. And thus, we do not make any claims about calcium-dependent conformational changes. We also want to note that due to high and consistent levels of calcium in the extracellular space, calcium is usually not used as a regulatory molecule outside of the cell although it is often important for structural stability of protein complexes (such as neuroligin and neurexin interaction).

The results of the experiment (treatment 45 hours post-transfection, 3 hours before signal readout) are shown in Figure X7, which show no significant differences between the PBS-treated and EGTA-treated samples, nor between the DMSO(0.1%)-treated and thapsigargin-treated samples.

	EV	zfGpr126 (-ss)	zfGpr126 (+ss)	§2AR
PBS vs. EGTA	ns	ns	ns	ns
DMSO vs. thapsigargin	ns	ns	ns	"

Figure X7. Effect of EGTA/thapsigargin on Gpr126 cAMP signaling.

(top, left) cAMP signaling levels for HEK293 cells transfected with empty vector (EV) and zebrafish Gpr126 splice isoforms treated with either PBS or 2mM EGTA 45 hours post-transfection (3 hours before measurement).

(top, right) cAMP signaling levels for HEK293 cells transfected with empty vector (EV) and zebrafish Gpr126 splice isoforms treated with either 0.1 μ M DMSO or 1 μ M thapsigargin 45 hours post-transfection (3 hours before measurement).

(bottom) Analysis of statistical significance: ns. $P > 0.05$; † $P < 0.05$; ‡ $P < 0.01$ ** $P < 0.001$ *** $P < 0.0001$: by two-way ANOVA and Tukey's multiple comparisons test.

Because we did not see a difference, we also tried an earlier treatment (21 hours post-transfection, 27 hours before signal readout) (Figure X8). In this experiment, we see significantly lower cAMP signal for both zfGpr126 -ss and +ss between PBS-treated and EGTA-treated samples as well as between DMSO-treated and thapsigargin-treated samples. This might suggest that calcium plays a role in the regulation of Gpr126 signaling. However, because we also see significant changes in signaling between PBS-treated and EGTA-treated §2AR as well as DMSO-treated and thapsigargin-treated §2AR it is likely that the EGTA and thapsigargin are affecting the cells in a non-specific manner since §2AR is not known to bind to calcium or mediate calcium-dependent signaling. In addition, we observed that the cells treated with either EGTA or thapsigargin looked unhealthy (rounder semi-detached) compared to the seemingly healthy cells that were treated with PBS or DMSO. We think that the EGTA and thapsigargin treatments are toxic to the cells and this may contribute to the observed lower cAMP levels. Altogether, we conclude that any effects of EGTA or thapsigargin on Gpr126 signaling cannot be definitively due to removal of calcium from the calcium-binding sites of Gpr126 receptors. And thus, we make no claims about the effect of calcium-dependent conformational changes.

We have modified the manuscript and removed our claims and changed the model in Figure 7. We think this wiki address the reviewer s concerns.

The authors responded: 'We also want to note that due to high and consistent levels of calcium in the extracellular space calcium is usually not used as a regulatory molecule outside of the cell although it is often important for structural stability of protein complexes (such as neuroligin and nerirexin interaction).'

I disagree on the lack of regulation through calcium as an extracellular first messenger in addition to its co-requirement for multimerization processes between protein partners. There even exist calcium sensing receptors. also GPCRs. that seem solely dedicated to responding to changes in extracellular calcium levels (see here: <https://doi.org/10.1038/366575a0>).

that ... p-p/

- The authors should mention and discuss why the tethered agonist peptide used in their Gpr126 stimulation assay differs significantly in size from the previously published one in Liebscher et al., 2014. The authors here used a p7 peptide, while in the original paper a p16 was identified as the most potent one from a library of gradually elongated tethered agonist portions and used to stimulate activity of Gpr126. This seems important as the p7 peptide reported in Liebscher et al. did not affect cAMP accumulation over control conditions. Have the authors tested a p16 peptide as well, what was the outcome compared to the p7 variant?

4 We thank the reviewer for bringing up this important point. We believe that there is a difference in peptide sensitivity between the zebrafish and human receptors, as well as differences in constructs used in our study compared to (Liebscher et al., 2014)

In (Liebscher et al., 2014) human GPR126 is used in the signaling assays. In the experiment where the authors tested peptides of increasing length on P2Y₁₂-A-GPS-CTF (a truncated and chimeric GPR126 construct), addition of 1 mM p7 increased cAMP levels 2-fold over untreated (Figure X9A). Addition of 1 mM of the longer p16 peptide increased cAMP levels over the untreated sample by 8-fold (Figure X9A). They also show that, using wild-type GPR126 treatment with 1 mM p16 increased cAMP levels 10-fold (Figure X9B)

(Liebscher et al., 2014)
 Figure X9. GPR126 Agonistic Peptides Are Derived from the C-Terminal Part of the GPS. (A) Application of 1 mM peptides of different lengths derived from the C-terminal part of the GPS, beginning at the cleavage site of GPR126, revealed agonistic properties as measured by cAMP accumulation. The highest agonistic efficacy was detected for a peptide containing 16 aa (p16). Negative controls: eV and GPR126 P2Y₁₂-DGPS-CTF mutant. Basal cAMP levels were 3.8 ± 1.6 nM.

(B) Different p16 concentrations were tested on WT P2Y12 WT GPR126 and P2Y12-DGPS-CTF. Inset: the concentration-response curve of p16 at WT GPR126 revealed an EC₅₀ > 400 nM. Basal eV levels were 3.2 ± 0.7 nM

In our signaling assay (Figure X10) we see that both isoforms of zebrafish Gpr126 are activated by both p7 and p14 peptides. Zebrafish Gpr126 is activated only slightly more by 1 mM p14 compared to 100 μM p7 which is why we chose to use 100 μM p7 in our activation assay. However, human GPR126 is only activated by p14 (~10-fold) and not p7. This could be due to an inherent difference between zebrafish Gpr126 and human GPR126 transmembrane domains which respond differently to different lengths of peptide. However, since (Liescher et al. 2014) showed activation of the truncated P2Y1t-AGPS-CTF GPR126 construct by shorter peptides, another possibility is that the human GPR126 transmembrane domain is blocked by its ECR, more so than the zebrafish version. This would prevent access of the p7 peptide on the full-length human GPR126. In this case activation would be observed only when the more potent p14 D16 peptide is applied.

We have modified Supplementary Figure 4C to show activation by p14 instead of p7 and have included the data for human GPR126 constructs as well.

Figure X10. Effect of p7 and p14 synthetic peptide on Gpr126/GPR126 cAMP signaling. cAMP basal signaling levels for HEK293 cells transfected with empty vector (EV), zebrafish Gpr126 splice isoforms, and human GPR126 splice isoforms treated with 2% DMSO, 100 μM p7 or 1 mM p14 15 minutes before measurement.

Thank you. This is an interesting piece of data and I'm glad it is now displayed in the revised manuscript.

p16, p390-392

The statement with regards to ligand binding ('the importance of furin cleavage is likely not primarily important for proper expression and trafficking but rather for a ligand-mediated function') has no experimental foundation. I would request to remove the statement regarding a potential ligand interaction through the SEA domain, or ask the authors to demonstrate evidence for ligand binding at the SEA domain in relation to furin processing (remove furin activity, remove furin cleavage site).

→ We agree that this is speculative and have removed the last part of this sentence.

Thank you.

Fig. S6

The authors mention that they engineered a furin-cleavage resistant Gpr126 mutant but show no evidence for this effect on furin cleavage. The authors are asked to demonstrate cleavage-deficiency effected by the mutation, e.g. by a Western blot.

I assume that R468A represents this mutation? If so, this is neither mentioned in the main text nor the figure caption, but only in the diagrams in S6B-D. Insert a suitable explanation in main text and caption.

→ R468A indeed represents the furin-cleavage mutant. We have included protein gels in Supplementary Figure 6 that show that ~50% of wild-type human GPR126 -ss and +ss isoforms are cleaved and that the R468A mutants completely abolish furin cleavage in both -ss and +ss isoforms. We also note that this mutation was first shown to be furin-cleavage resistant in a previous study (Moriguchi et al., 2004) and have clarified this in the manuscript. In addition, we have removed Supplementary Figures 5C-D due to unreliable results but have provided explanations of S5B (now S5E) in the main text and caption.

'We have removed Supplementary Figures 6C-D due to unreliable results but have provided explanations of S6B (now S6E) in the main text and caption.' (I have corrected the numbers in this citation from the rebuttal letter, since they were incorrectly labeled by the authors)

OK. Since the omitted datasets were never referred to in the original submission at all, they likely did not figure prominently in the authors' interpretation of the role of the furin cleavage on Gpr126 signaling.

MINOR POINTS

p3, 67-68

There are only 32 human aGPCR genes currently known, EMR4 is considered a pseudogene in

the human genome. Please correct.

→ We have corrected this number throughout the manuscript.

p5, 120-121

'... 150 aa region between PTX and HormR, could not be identified.'

I'd suggest to rephrase since apparently the fact that there is a linker region between the said domains, to which a structure could not be readily assigned, was already known.

'... 150 aa region between PTX and HormR, could not be assigned to a known structural fold.'

→ We have changed this sentence accordingly. However, we note that the presence of the linker region was not known prior to the current study.

Figure S4E

A grouping of observed conformations such as presented in Fig. 3D is missing for this dataset. Please provide.

→ We have now included the grouped class averages (now Supplementary Figure 4D).

Figure caption S4C

'Basal signaling for sam constructs ...'. Correct

→ We have corrected this typo.

p15, 370

Insert bold words in statement: 'Although THE furing cleavage SITE is not conserved.....'

→ We have modified this sentence accordingly.

p16, 388-389

There is a problem with grammar of this sentence:

'HEK293 cells transfected with a mutant form of human GPR126 that is not able to be cleaved by furin were detected on the cell surface. '

Change to:

'Human GPR126 that is not able to be cleaved by furin was transfected in HEK293 cells and detected on the cell surface. '

→ We have modified this sentence accordingly.

Fig. 7E

Is the ECR shed after furin cleavage? There is not evidence for this in this manuscript, and this would also be in contrast to data obtained from the Notch receptor field showing that furin cleavage results in a non-covalent heterodimer (similar the SEA and GAIN domain autoproteolysis), which does not readily dissociate (see Logeat et al., PNAS, 1998; Blaumueller et al., Cell 1997, Rand et al., Mol Cell Biol, 2000).

Hence, the cartoon introduces a situation - NTF shedding - that is has not been formally experimentally associated with neither furin cleavage, SEA domain autoproteolysis nor GAIN domain self-cleavage. The latter point has been only partially addressed by artificially or genetically produced Gpr126 NTF, which can rescue Gpr126 hypomorphic phenotypes in vivo. However, to my knowledge it was not tested whether these phenotypes in heart development

can be complemented by using a GAIN domain autoproteolysis-disrupting mutation in Gpr126. In sum, the scenario of NTF shedding is very speculative and further elaboration on this without concrete evidence is misleading readers.

→ We agree that GPR126 furin cleavage probably acts similarly to cleaved SEA domains from Notch in that cleavage does not immediately lead to shedding, and this is supported by Supplementary Figure 6B showing elution of furin-cleaved ECR fragments together from a size exclusion column at the same volume as the full-length uncleaved ECR. However, we agree that the shedding model is speculative, and we have modified this figure to simplify it and to only show scenarios for which we have provided evidence.

OK.

Kuffer, A., Lakkaraju, A.K., Mogha, A., Petersen, S.C., Airich, K., Doucerain, C., Marpakwar, R., Bakirci, P., Senatore, A., Monnard, A., *et al.* (2016). The prion protein is an agonistic ligand of the G protein-coupled receptor Adgrg6. *Nature* 536, 464-468.

Liebscher, I., Schon, J., Petersen, S.C., Fischer, L., Auerbach, N., Demberg, L.M., Mogha, A., Coster, M., Simon, K.U., Rothmund, S., *et al.* (2014). A tethered agonist within the ectodomain activates the adhesion G protein-coupled receptors GPR126 and GPR133. *Cell reports* 9, 2018-2026.

Moriguchi, T., Haraguchi, K., Ueda, N., Okada, M., Furuya, T., and Akiyama, T. (2004). DREG, a developmentally regulated G protein-coupled receptor containing two conserved proteolytic cleavage sites. *Genes to cells : devoted to molecular & cellular mechanisms* 9, 549-560.

Paavola, K.J., Sidik, H., Zuchero, J.B., Eckart, M., and Talbot, W.S. (2014). Type IV collagen is an activating ligand for the adhesion G protein-coupled receptor GPR126. *Science signaling* 7, ra76.

Petersen, S.C., Luo, R., Liebscher, I., Giera, S., Jeong, S.J., Mogha, A., Ghidinelli, M., Feltri, M.L., Schoneberg, T., Piao, X., *et al.* (2015). The adhesion GPCR GPR126 has distinct, domain-dependent functions in Schwann cell development mediated by interaction with laminin-211. *Neuron* 85, 755-769.

Response to Reviewer Comments

Structural basis for adhesion G protein-coupled receptor Gpr126 function
NCOMMS-19-13915A

Reviewer #1 (Remarks to the Author):

The authors have dealt with all the comments in my review satisfactorily.

→ We thank Reviewer #1.

Reviewer #2 (Remarks to the Author):

The authors have answered all of my concerns.

→ We thank Reviewer #2.

Reviewer #3 (Remarks to the Author):

The authors have made the necessary changes and text edits and I am satisfied with the state of the paper. This is a good solid piece of work.

→ We thank Reviewer #3.

Reviewer #4 (Remarks to the Author):

I thank the authors for the meticulous revision of their paper, which has substantially improved. Regarding the rectification of uncertainties pertaining to the interpretation of datasets I am satisfied. In order to keep track of the extensive changes that followed upon my inquiries I listed all points that the authors dropped since their initial claims could not be confirmed after further experimentation, which I like to list here again also for the consideration of the other referees and the editor:

Re Calcium-binding site/structural changes/signaling interconnection

- 'We are aware that the simple model contradicts the data; and that the more complex and ill-explained model agrees with the data. For this reason, we prefer to move the mutant data to Supplementary Figure 4H and remove claims about it and do not mention the complex model.'

Re Ligand-binding at the calcium binding site:

- 'Because of the lack of evidence for the calcium-binding site as a ligand-binding site, we have modified the manuscript and the model in Figure 7 to remove claims that are not supported by evidence.'

- 'We have tried to reproduce the results that show that type IV collagen and PrPC bind to Gpr126, as well as show a difference in ligand-binding between the -ss and +ss isoforms, but our results were inconclusive.'

- 'We observed no effect of ligands on receptor signaling. We make no claims in the manuscript.'

Re Calcium-dependent conformational changes:

- 'We do not make any claims about calcium-dependent conformational changes.' Re Furin cleavage and signaling:

- Removal of Supplementary Figures 6C-D due to unreliable results

I like to repeat that I think that structural and in vivo data presented here are reliable. However, I would also like to point out that pharmacological data do not yet connect these concepts coherently. After the revision it seems clear that, unsurprisingly, only splicing affects signaling outcome of Gpr126 as a means of transcriptional control of Gpr126 function (see here for publications that have demonstrated this recently using other adhesion GPCRs: 10.1016/j.celrep.2019.01.040, 10.1038/s41598-019-46265-x), but neither calcium, calcium binding, or ligand binding to the calcium binding site as post-translational means do.

This regretful situation notwithstanding, I believe that the novel structure of the complete ECR of Gpr126 is interesting and deserves publication in Nat Commun.

I would ask that a few remarks, largely pertaining to negative data that accumulated throughout the revision process, be mentioned in the main text in order to avoid the confusion for the reader that I faced when first reading the manuscript. Comments on this are inserted below next to the responses of the authors. I do not need to see the final revision of the manuscript again if the editorial office can ensure that the points were included in the manuscript.

→ We thank the reviewer for the thoughtful response to our revisions. We agree that modification of the text would make our work clearer to the reader. We have incorporated the changes to the manuscript accordingly.

(Re: Error bars for expression data)

OK. Please include a brief version of explanation in the caption as to why no error bars are shown for the surface expression value in Fig. 4A, S4B and S6B, i.e. the fact that shown is only a single flow cytometry measurement. I am still uncertain why the authors do not show the pooled instead of the current representative graph of all the three separate replicates they conducted as they indicate in the rebuttal letter. From what is apparent in Fig. X3 this should not affect the general outcome of this assay but would offer the reader a more generalizable view of this dataset.

→ We have included an explanation in the figure legends for Figure 4, Supplementary Figures 4 and 6. We are inclined to show single experiments rather than pooled experiments because the luminescence measurements can vary between experiments depending on cell passage number or different lots of reagents. We observe consistent signaling trends between constructs, but the raw values for the constructs may not always be consistent.

(Re: Calcium-binding site/structural changes/signaling interconnection)

I thank to authors for reassessing this part of the manuscript and toning down their claims drawn from the collective assessment of structural and pharmacological experiments in this paper. However, in the main manuscript it still reads:

'These results suggest that the calcium-binding site is important for increased signaling in the (+ss) isoform and may be a functional site for Gpr126.'

It is not made clear in the text and even more confusing to the reader that structural, functional and calcium-binding site mutational data are at a mismatch with each other here, especially after moving the mutation dataset out of the direct view of readers to the supplement. If the authors 'agree that these sets of signaling data are confusing and seem to be contradictory to our argument that the calcium-binding site is important for cAMP signaling'.

Please say so at this point in the main manuscript in the summary sentence of this results paragraph.

At the moment, there is a logic connection insinuated by the authors that cannot be substantiated. I see no flaw in actively making the readers aware of this disconnect, but argue that this makes the main messages of the manuscript even more clearer, credible and stimulating. So please comment on this in the results and discussion.

The authors have delivered a meticulous discussion of alternative explanations in the rebuttal letter, which I suggest to condense and transfer to the manuscript's discussion on this matter.

→ We thank the reviewer for these suggestions, and we have made changes in the results and discussion sections to make the reader more aware of the confusion here and our explanation of the results.

(Re: Ligand binding to Gpr126/GPR126)

I thank for authors for their efforts and agree with the authors that that the experiments on Gpr126-PrP-C-FT and collagen IV interactions 'at their current state, fail to demonstrate a reliable direct interaction of Gpr126/GPR126 with the tested ligands.' I also commend the authors on being frank about the failure to repeat these experiments that were previously highly visibly published (Paavola et al., Sci Signal 2014; Kuffer et al., Nature, 2016).

→ We thank the reviewer for acknowledging our efforts despite the inconclusive results.

(Re: Altering calcium in signaling assay)

The authors responded: 'We also want to note that due to high and consistent levels of calcium in the extracellular space, calcium is usually not used as a regulatory molecule outside of the cell although it is often important for structural integrity of protein complexes (such as neuroligin and neurexin interaction).'

I disagree on the lack of regulation through calcium as an extracellular first messenger in addition to its co-requirement for multimerization processes between protein partners. There even exist calcium sensing receptors, also GPCRs, that seem solely dedicated to responding to changes in extracellular calcium levels (see here: <https://doi.org/10.1038/366575a0>).

I agree that the conducted experiments with EGTA and thapsigargin do not allow to conclude that that calcium binding site and calcium binding affect signaling of Gpr126.

→ We thank the reviewer for the correction regarding GPCRs regulated by extracellular calcium and for agreeing about the inconclusive results of these assays.

(Re: Signaling activation assay)

Thank you. This is an interesting piece of data and I'm glad it is now displayed in the revised manuscript.

→ We thank the reviewer for raising the important point which led to the revision of this figure.

(Re: Furin cleavage)

'We have removed Supplementary Figures 6C-D due to unreliable results but have provided explanations of S6B (now S6E) in the main text and caption.' (I have corrected the numbers in this citation from the rebuttal letter, since they were incorrectly labeled by the authors)

OK. Since the omitted datasets were never referred to in the original submission at all, they likely did not figure prominently in the authors' interpretation of the role of the furin cleavage on Gpr126 signaling.

→ We agree with the reviewer on this point and would like to explore the role of furin cleavage on Gpr126 signaling in future experiments.